# Assessment of the Fugitive Emission Distributed Sampling (FEDS) system: A mobile, multi-inlet system for continuous emissions monitoring

Jacob T. Shaw[1], Neil Howes[1], Jessica Connolly[1], Dragos E. Buculei[1], Jamie Ryan[1], Jon Helmore[1], Nigel Yarrow[1], David Butterfield[1], Fabrizio Innocenti[1], Rod Robinson[1]

[1]National Physical Laboratory, Hampton Road, Teddington, Middlesex, TW11 0LW, UK.

*Correspondence to*: Rod Robinson (rod.robinson@npl.co.uk)

**Abstract.** The National Physical Laboratory (NPL) has developed and trialled a mobile and remotely-operated Fugitive Emission Distributed Sampling (FEDS) system for continuous measurements of emissions at the facility spatial scale. FEDS is capable of both locating and quantifying emission sources over long-term periods and has been deployed at sites around the UK to monitor methane emissions from the natural gas network, landfill, and waste treatment. This work presents assessment activities using a controlled release facility (CRF) to test the performance of the measurement system and two reverse dispersion models (Airviro and WindTrax) for emission quantification. The CRF was used to simulate a simple single-point methane emission source with constant release rate of 1 kg h$^{-1}$ over four separate experiments. Emissions were quantified using prior knowledge of release timings as well as in the absence of this knowledge. High variability in wind direction was shown to negatively impact emission quantification accuracy (especially for Airviro). Emission results were improved by removing periods of high wind variability (low wind persistence) from analysis. Both models performed better when using daily-averaging periods for emissions (Airviro RMSE = 0.37 kg h$^{-1}$ (37% relative); WindTrax = 0.29 kg h$^{-1}$ (29%)) over shorter averaging periods, such as hourly data (Airviro RMSE = 0.77 kg h$^{-1}$ (77%); WindTrax = 2.19 kg h$^{-1}$ (219%)). Estimated emission rates were shown to be sensitive to the specified source release height for both models, with discrepancies in model release height relative to the true release height of more than 0.5 m yielding less accurate results.

**Keywords**: Methane, emission, network, dispersion, validation, continuous, long-term

# 1 Introduction

There is growing awareness of the need to reduce global anthropogenic emissions of methane ($CH_4$), a powerful greenhouse gas, to meet the goals of the Paris Agreement (Nisbet et al., 2020; 2022a; Ocko et al., 2021; Tollerson, 2022). The Global Methane Pledge, signed by over 100 nations at the 26[th] United Nations Climate Change Conference of the Parties (COP26) in 2021, requires signatories to cut their $CH_4$ emissions by 30% by 2030 (European Commission, 2021). Yet only 13% of global $CH_4$ emissions are currently included within direct mitigation policies (Olczak et al., 2023). $CH_4$ mitigation requires efficient, accurate, and transparent monitoring of $CH_4$ emissions, both to ensure effective reductions are taking place and to ensure public confidence in strategies (Erland et al., 2022; Connor et al., 2024). However, monitoring $CH_4$ emissions is extremely challenging, especially across the diverse spatial, temporal, and emission magnitude scales required to guarantee accuracy and reliability, and to fully capture all known and unknown emission sources (Cusworth et al., 2020; Nisbet, 2022b; Wang et al., 2022).

Connor et al. (2024) presented a framework for monitoring $CH_4$ emissions highlighting the complexity of the monitoring landscape. Connor et al. (2024) emphasised that monitoring methods should adhere to basic principles of metrology and quality assurance in order to meet their data reporting requirements. Where possible, standardised methods should be used for formal reporting of $CH_4$ emissions but validation activities in controlled scenarios are an important aspect of achieving trust and confidence in any developing (or even standardised) method. Measurement-based approaches to $CH_4$ emission reporting are now being prioritised over (or alongside) site-level estimation via activity data through programmes such as the Oil and Gas Methane Partnership 2.0 (OGMP2.0) and within recent regulatory requirements in the European Union (i.e., EU 2024/1787, 2024). OGMP2.0 requires transparent and verified site-level measurements to achieve its Gold Standard whilst EU 2024/1787 requires mandatory measurement, reporting, and third-party verification of site-level emissions.

Accurate monitoring of facility-scale $CH_4$ emissions typically requires measurements of atmospheric $CH_4$ around the facility of interest. There are many approaches to sampling atmospheric $CH_4$, whether that be from single-point stationary sensors, from a network of sensors, or from sensors mounted on a plethora of mobile platform options, including people, bicycles, ground-based vehicles, unmanned aerial vehicles (UAVs), manned aircraft, or satellites (e.g., Fox et al., 2019; Shaw et al., 2021; Erland et al., 2022; Jacob et al., 2022; Allen, 2023; Yang et al., 2023; Connor et al., 2024). Mobile platforms offer advantages in spatial coverage and may be used to target individual emissions sources for direct sampling. However, using mobile platforms can be labour intensive and therefore long-term or continuous deployment can be challenging. The labour-intensity of performing mobile measurements means options such as the tracer gas dispersion method (e.g., Mönster et al., 2019; Shah et al., 2025a) or those utilising unmanned aerial vehicles (e.g., Shaw et al., 2021) are typically used for so-called "snapshot" monitoring and may therefore miss intermittent emission sources (if not measuring at the time of emission) or have limited capacity for determining variability in emissions over time (Connor et al., 2024). On the other hand, stationary sampling can provide capacity for reduced labour efforts (as sensors can be left in place) with the benefit of potential for continuous monitoring. Where single-point stationary sampling may have limited spatial coverage this can be alleviated by

using a network of stationary sensors. However, installing networks for continuous monitoring can be prohibitive due to the expense of high precision analysers and a lack of reliable and high-performance low-cost alternatives (Fox et al., 2019).

Kumar et al. (2022) presented results from several controlled-release validation experiments on a $CH_4$ monitoring network consisting of seven high-precision gas analysers (of various types) connected to 16 sampling inlets. Using multiple spectroscopic analysers allowed for the potential to monitor both the emission plume and the $CH_4$ background simultaneously (regardless of wind direction). However, using multiple expensive analysers would typically be beyond the scope of routine emission monitoring especially in the context of the many facilities in operation nationally and globally. Riddick et al. (2022a) used a network of four calibrated low-cost sensors positioned 30 m from emissions of known rates over two days. The low-cost sensors were able to detect $CH_4$ emissions of 84 g h$^{-1}$ for only 40% of the time. Emissions estimated using reverse dispersion models were found to be only within ±50% of the release rate on a 1-day average. In addition, there are concerns that low-cost $CH_4$ sensors may be unreliable due to cross interference with parameters other than $CH_4$ (Riddick et al., 2022; 2020; Shah et al., 2025b) and may only be applicable in limited environmental conditions (Rivera-Martinez et al., 2024). Furthermore, in-field calibration of low-cost sensors remains challenging; low-cost sensor calibration is typically performed in chamber experiments prior to or following, but not during, deployment (Lin et al., 2023). Routine calibration of multiple in-field low-cost sensors, even without a chamber, would also be labour intensive and non-optimal.

An alternative solution to using multiple expensive spectroscopic analysers or a network of (currently) imprecise and unreliable low-cost sensors is to use a single expensive analyser connected to multiple sampling locations. The use of multiple sampling locations gives a few direct advantages over single-point monitoring. Firstly, multiple points-of-interest may be monitored in quick succession; separate emission sources may therefore be disaggregated from greater spatial data. Secondly, multi-point monitoring is less dependent on wind direction (and wind speed) whereas a single-point monitoring system would be reliant on a potentially narrow range of wind directions across which emissions can be captured; wind directions outside of this range might result in measuring air masses which have not travelled over the source of interest (Shaw et al., 2020). Single-point monitoring is therefore much more likely to miss transient and irregular emission events from non-continuous sources due to imperfect wind conditions. With all fence-line monitoring approaches, and even with multiple sampling locations, there still remains the possibility of missing emissions. Emissions may be missed, for example, if gaseous emissions are transported above the height of the sampling inlets, which can occur if the emission source emits gases at a higher temperature than the ambient atmosphere. This may be mitigated, of course, by placing inlets at multiple heights above the ground.

In this work, we evaluate the performance of the Fugitive Emission Distributed Sampling (FEDS) system, developed at NPL, for detecting, locating, and quantifying $CH_4$ emission sources. Multiple known and controlled releases of $CH_4$ (non-blind, single point emission source) were used to evaluate and assess the FEDS performance. The FEDS system is intended for the continuous but cost-effective monitoring of $CH_4$ emissions. Furthermore, FEDS is envisioned to be accessible to non-academic users, and this is exemplified through the use of commercially-available software for the direct quantification of $CH_4$ emissions. We also present results from a custom-built Gaussian plume model which provides additional context for

assessment of emission quantification performance. Procedures for the quality assurance of local-scale reverse dispersion data

were examined, with emphasis on the impact of wind conditions on measured emission rates.

## 2 Methodology

The FEDS system uses a mobile weatherproof station housing a high-performance gas analyser (see Appendix A) for the measurement of atmospheric trace gas concentrations. In principle, any high-performance analyser with sufficient capability for high resolution measurement of trace gases may be used. For the purposes of the work described here, an ABB Ltd. Los Gatos Research Inc. (LGR) Fast Methane-Ethane Analyzer (FMEA; GLA331 series; see Appendix A for specifications) was used to measure atmospheric methane ($CH_4$) and ethane mole fraction data. A multiple-inlet unit (MIU; or multiplexer) is used to switch between up to 15 sampling inlets, located strategically around a facility to both optimise proximity to known emission sources and to capture emissions across a full range of possible wind directions (i.e., covering 360° around the target facility). Each inlet location is sampled in a pre-determined sequence providing discrete atmospheric trace gas measurements at high precision (i.e., nmol mol$^{-1}$ (SI-preferred), or parts-per-billion (ppb; used throughout this work) for $CH_4$) across a wide spatial area at least once every hour. In the current setup, a sixteenth sampling inlet is reserved for a single-point reference gas standard containing a known concentration of $CH_4$. Sampling inlets were protected from the weather via solar radiation shields attached at heights of ~2 m above the ground. Synflex 1300 tubing (polyethylene/aluminium composite) was used to connect sampling inlets to an external box on the FEDS trailer. Internally (within the trailer) all tubing was polytetrafluoroethylene (PTFE). Groups of four sample lines were connected to one of four vacuum pumps, each with a specified throughput of 20 L min$^{-1}$. This created a draw of up to 5 L min$^{-1}$ (realistically 2-3 L min$^{-1}$) at each sampling inlet location, depending on the relative distances of the four tubing lengths. The MIU was connected to each sampling inlet individually, as well as to the reference gas standard.

The central station also contains a 10 m telescopic hydraulic mast for the measurement of local meteorology (wind speed, wind direction, and air temperature). Meteorological data is also collected from the trailer roof to acquire air temperature data at a second height. The acquisition of trace gas and meteorological data can be controlled and monitored remotely. Meteorology data was recorded by a CR6 datalogger (Campbell Scientific). Both the LGR FMEA and the CR6 datalogger were networked via a 3G cellular router for remote access. The 3G network in the UK was sufficient for cellular reception for the duration of the measurements in this work (although the router has since been upgraded for use with the 4G network).

Figure 1 shows a schematic of the FEDS facility used in this work. Measurements for the work presented here took place between 21st March and 8th April 2022.

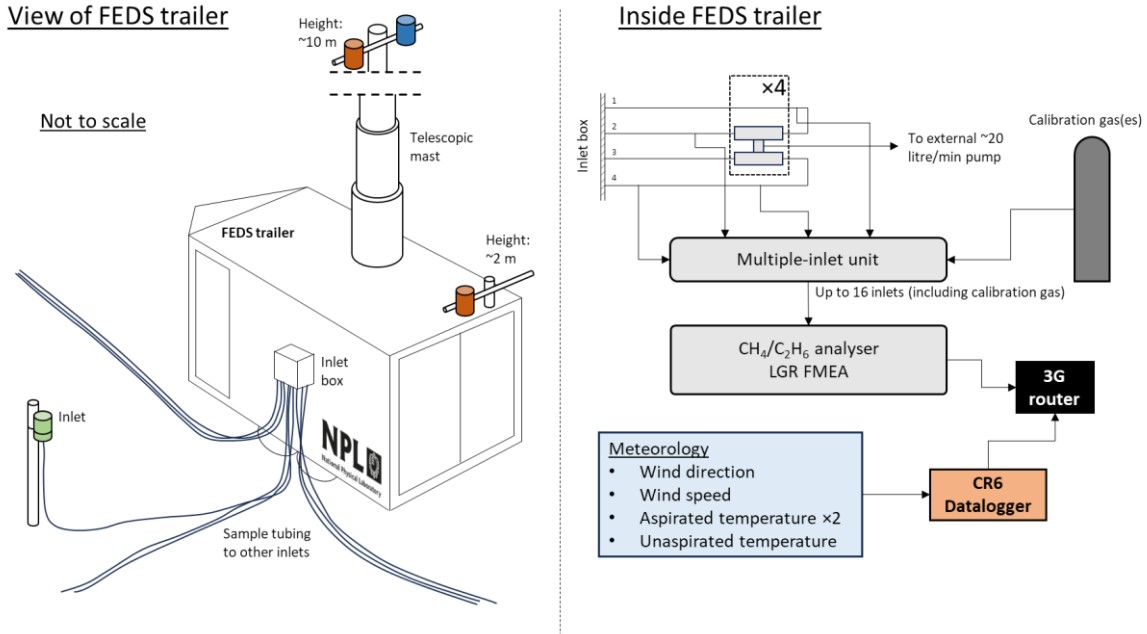

**Figure 1. Schematic showing the FEDS facility. LGR FMEA = Los Gatos Research Inc. Fast Methane-Ethane Analyzer. The heights on the FEDS trailer are defined as heights above the ground.**

## 2.1 Methane mole fraction measurements

For the work presented here, an LGR FMEA was used to measure mole fractions of atmospheric $CH_4$ between 21st March and 8th April 2022. Atmospheric ethane ($C_2H_6$) mole fraction measurements were also recorded by the FMEA but $C_2H_6$ data were

125 not assessed in this work. The FMEA has a reported precision of 2 ppb $CH_4$ (1σ) at 1 s and 0.6 ppb $CH_4$ (1σ) at 10 s, over a measurement range of 0 – 10,000 µmol mol$^{-1}$ (or parts-per-million; ppm) $CH_4$ (Table 1). Whilst the FMEA normally operated at 1 Hz, a software issue led to a slow and steady degradation of sampling frequency over time (tests showed that measurements were made at roughly 0.25 Hz after approximately two weeks, and at roughly 0.15 Hz after approximately five weeks). Periodic cycling of the instrument power temporarily resolved the software issue and allowed the analyser to return to a sampling

frequency of 1 Hz albeit with the same degradation pattern. The software issue was not expected to have impacted the accuracy of measured $CH_4$, although the reduced number of datapoints taken at each sampling location increased the associated standard error. The time taken for gas to be flushed through the optical cell (cell turnover time) was ~20 s; this turnover time is similar to those reported for analogous but smaller gas analysers (e.g., France et al., 2021).

The MIU switched between sampling lines every 4 minutes so that $CH_4$ measurements were taken from each of the

135 15 inlet locations at least once every hour. Even whilst inactive on the MIU, the pumps were used to flush air through the sampling lines to maintain constant flow. However, the first period of data following switching to a new sampling line were considered invalid to remove the possibility of the FMEA measuring air drawn from the previous sampling line (e.g., due to non-instantaneous cell flushing time). The sensitivity of quantified emissions to this time period (referred to as the sampling

lag time) was tested to determine the optimal time period to invalidate or exclude data (see Appendix C). Mole fraction data from each sampling location were averaged over hourly intervals to provide one single value per sampling location per hour to the emission quantification model(s).

A single reference gas standard (BOC Ltd.) with a nominal $CH_4$ mole fraction of 50.0 ($\pm$ 0.5) μmol mol$^{-1}$ was sampled by the gas analyser periodically as a quality check to ensure traceability of measurements. The gas standard was sampled once per day on each of 24$^{th}$, 28$^{th}$, 29$^{th}$ and 31$^{st}$ March and 1$^{st}$ and 5$^{th}$ April, and twice on each of 30$^{th}$ March and 6$^{th}$ April, for four minutes each time. The first 60 seconds of data were removed (as above when switching over sampling inlets) to preclude the possibility of measuring air in the cell sampled from elsewhere (i.e., one of the other sampling inlet locations). The mean $CH_4$ mole fraction measured when sampling from the gas standard was 50.22 ($\pm$ 0.04) μmol mol$^{-1}$ across all samples, with a median value of 50.23 μmol mol$^{-1}$ and a range 50.11 to 50.29 μmol mol$^{-1}$. Analyser settings were not adjusted following sampling of the gas standard as all of the measured mole fractions were well within the uncertainty range of the reference gas standard. In practice, it would be preferential to employ a calibration scheme with more than one single standard reference point to account for any non-linearity in analyser response across a range of $CH_4$ mole fractions. However, the consistency between the analyser response and the reference gas standard provides enough confidence that the data was of high precision for the duration of this field campaign.

$CH_4$ mixing ratios measured whilst the flow into the optical cell was less than zero (which occurred during initial instrument start up and infrequently as sampling inlets were switched) and when the cell pressure was below 298.5 Torr (the target cell pressure) were removed before analysis due to inaccurate readings (i.e., mixing ratios well below the atmospheric $CH_4$ background).

## 2.2 Meteorology

Wind speed and wind direction were measured using a WindSonic anemometer (Gill Instruments) deployed at 10 m above ground on the hydraulic mast. Manufacturer-reported specifications are detailed in Table 1. Atmospheric temperature was measured at two heights using PT100 platinum temperature probes (R. M. Young Company). Temperature was measured at 2 m above ground in both an aspirated and unaspirated radiation shield and at 10 m above ground in an aspirated radiation shield only. The PT100 temperature probes were calibrated against a National Institute of Standards and Technology (NIST) traceable thermometer by R. M. Young Company on 6$^{th}$ August 2020. All meteorological data were recorded as one-minute average data. Meteorology data were averaged across hourly periods for input into the emission quantification model(s).

**Table 1. Manufacturer-reported specification for measured parameters.**

| Parameter | Accuracy | Resolution | Response time |
|---|---|---|---|
| Wind speed | ±2% @ 12 m s$^{-1}$ | 0.01 m s$^{-1}$ | 0.25 seconds |
| Wind direction | ±3° @ 12 m s$^{-1}$ [a] | 1° | 0.25 seconds |
| Temperature | ±0.3 °C @ 0 °C<br>±0.1 °C with NIST calibration | | |
| Methane ($CH_4$) | 2 ppb (1σ) @ 1 s<br>0.6 ppb (1σ) @ 10 s | 0.01 ppb | 1 Hz [b] |

[a] Additional uncertainty in wind data is anticipated as the anemometer was only roughly aligned to the north (visually using a compass) and hence the accuracy of wind direction data quoted here should be treated as a minimum value only.

[b] A software issue on the FMEA led to the steady degradation of sampling frequency of $CH_4$ mole fractions over time. Periodic recycling of instrument power resolved this issue, but reduced data collection did occur.

The stability of local wind conditions can affect the accurate measurement of emissions and derivation of emission rates, particularly when averaging concentration measurements over periods of time during which the wind conditions were variable. This is partially because the average wind direction over a period of time may not be representative of the true range in wind directions during that period. For example, 30 minutes of wind from the east (90°) and 30 minutes of wind from the south (180°) would yield an hourly-average wind direction from the south southeast (135°). In reality, the wind did not blow from the southeast, and enhancements in methane concentration may only have been measured at sampling locations directly to the west (when the wind was from the east) and to the north (when the wind was from the south) of the emission source, and not at locations to the northwest. To limit the reporting of lower quality emission data, a simple measure of wind persistence was calculated. The wind persistence was calculated by determining the ratio of the mean vector wind speed to the mean scalar wind speed (Eq. 1) (Shirvaikar, 1972). Wind persistence therefore varied between 0 and 1, with a wind persistence value of 1 indicating extremely stable winds, and a wind persistence value of 0 indicating extremely unstable (or highly variable) winds. In principle, wind persistence can be calculated for any time averaging period but was calculated here for hourly periods to match the averaging period for the meteorology and $CH_4$ mole fraction data.

$$Wind\ persistence = \frac{|\bar{z}|}{\bar{z}} \tag{1}$$

Where $|\bar{z}|$ is the mean vector wind speed, and $\bar{z}$ is the mean scalar wind speed, calculated via Eq. 2 and Eq. 3 respectively.

$$|\bar{z}| = \sqrt{\bar{u}^2 + \bar{v}^2} \tag{2}$$

$$\bar{z} = \overline{\sqrt{u^2 + v^2}} \tag{3}$$

Where $u$ and $v$ are the horizontal and vertical components of the wind respectively. The impact of wind persistence on the emissions quantified using different dispersion models was tested. The application of various thresholds for invalidating periods of low wind persistence was also tested.

## 2.3 Field trial setup

NPL developed and operate several controlled release facilities (CRFs; IGI Systems Ltd.) capable of releasing multiple gases
simultaneously from different release points to mimic 'real-world' emission scenarios. A detailed description of one such CRF is provided by Gardiner et al. (2017) although it should be noted that the CRF operated by Gardiner et al. was a different instrument to the CRF operated as part of this work. The CRFs can be, and have been, used for the validation of a range of gas detection and emission rate quantification solutions (Gardiner et al., 2017; Shah et al., 2019; EN17628:2022, 2022; Defratyka et al., 2025). In brief, the CRFs comprise an array of gas cylinders connected to a network of mass flow controllers (MFCs)
calibrated for flow rates across multiple orders of magnitude. The gas release point(s) can be located within approximately 50 m of the CRF equipment. A wide range of accurate release rates can be achieved by using different combinations of MFCs. The CRF used in this work provided accurate and precise releases of different gases; for $CH_4$, the CRF was capable of volumetric releases from 5 mL min$^{-1}$ to 500 L min$^{-1}$, equivalent to approximately 0.0002 kg h$^{-1}$ to 21 kg h$^{-1}$ with low uncertainty and metrological traceability (to the SI, or International System of Units). Only one release rate was tested during this
assessment campaign but there are plans to validate the FEDS system against a range of release rates in future campaigns.

       The FEDS field campaign was conducted over three weeks in March and April 2022. The CRF and FEDS systems were set up in a flat and open area, with no substantial structures or changes in elevation for more than 250 m in all directions, at Bedford Aerodrome (Bedfordshire, UK). For the assessment campaign, 12 inlet locations were arranged in a circular pattern around the centrally-located emission release point. Table 2 presents some of the pertinent details regarding the field campaign,
Table 3 some data regarding each release, and Figure 2 illustrates the CRF $CH_4$ release rates and the FEDS setup. For each release, methane was released continuously (day and night) for at least 20 hours; this temporal release pattern may be representative of continuous emission sources but less representative of short-term (transient) emission sources or emission sources which vary over time. Whilst the release rate was known prior to analysis, all data analysis was performed independent of the knowledge of the release rate (i.e., knowledge of the release was not used to influence analysis). A release height of ~2
220    m may be broadly representative of some low-level elevated sources but not of higher elevated sources such as stacks or chimneys. Similarly, the use of an elevated release height (and a single release location) is unlikely to well represent landfill emission sources where methane is typically emitted through the subsurface across a wide area. The release scenario used here was chosen to be reasonably simple to conduct a first assessment of the FEDS system.

**Table 2. Field campaign details.**

| Campaign | Field campaign 2022 |
|---|---|
| Date | 22nd March – 8th April 2022 |
| Location | Bedford Aerodrome |
| Number of sampling inlets | 12 |
| Sampling inlet heights | Fixed at ~2 m |
| Number of controlled releases | 4 |
| Duration of controlled releases | >20 hours |
| CRF $CH_4$ release heights | Fixed at ~2 m |
| CRF $CH_4$ release rates | 1 kg h$^{-1}$ |

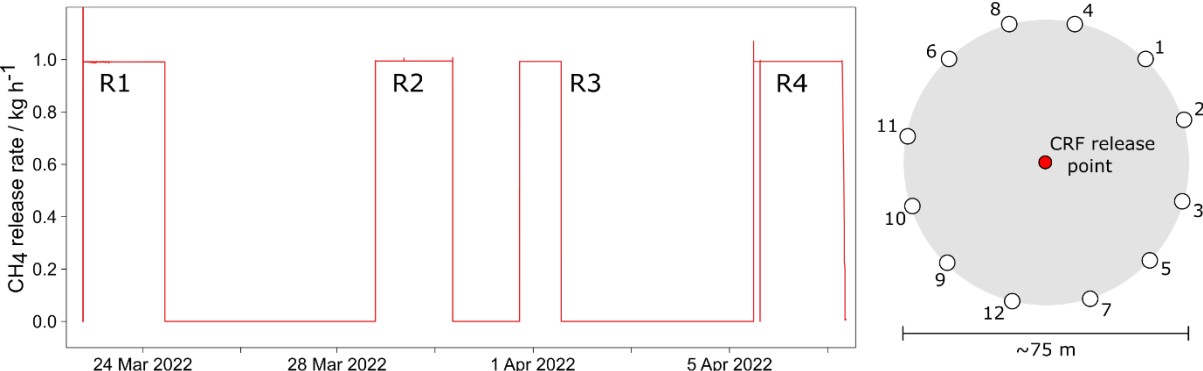

**Figure 2. CRF controlled release rates (left) for each of the four numbered controlled releases (R1 through R4) and FEDS inlet locations (right; numbered 1-12) relative to the CRF release point location (red) for the campaign (see Appendix A for photograph reference). See Table 3 for the mean $CH_4$ emission rate during each of the four releases. Note that inlet location numbers were primarily for the purposes of tracking metadata and were not indicative of sequential sampling order by the gas analyser (i.e., the sampling order was not in numerical order). Equivalent flows were achieved at each sampling inlet using different vacuum pumps.**

## 2.4 Emission quantification

The spatial scales involved in this work, and for which FEDS is intended (i.e., local and facility-scale) necessitates that any models used for quantifying emissions are capable of simulating atmospheric dispersion over these relatively small spatial scales (<1 km). Many popular inverse models may be more typically suited to modelling atmospheric dispersion over larger spatial scales (>1 km; e.g., the Numerical Atmospheric-dispersion Modelling Environment, NAME, or the FLEXible PARTicle dispersion model, FLEXPART) and were not tested here (Jones et al., 2007; Pisso et al., 2019). Three inverse dispersion models which have been used for facility-scale emissions quantification were used to quantify $CH_4$ emission rates here. These three models were chosen following an internal commercial review of available options (of both software and freeware) assessing model usability, availability, support, price, operating principle (Gaussian, Lagrangian etc.). At least one of each type of common approach to modelling atmospheric transport (Gaussian, Langrangian) was chosen so as to compare

the two methods. It should be noted that other modelling options are available but testing all available models was beyond the scope of this work. The three models used included:

- Airviro (v4.0; Apertum, www.airviro.com), an air quality management system with an inverse Gaussian plume dispersion model (Wickström et al., 2010).
- WindTrax 2.0 (ThunderBeach Scientific, www.thunderbeachscientific.com), a Lagrangian stochastic particle model used for modelling atmospheric transport over small horizontal distances (<1 km) (Flesch et al., 1995; 2009).
- A custom-built Gaussian plume model, similar in concept to that described in OTM-33A (US EPA, 2014).

The three models, and their set up, are described in more detail in Appendix B.

Two different approaches to quantifying emissions were tested. Firstly, emissions were quantified using knowledge of the precise $CH_4$ controlled release timings. This was done by creating subsets of $CH_4$ and meteorological data from which emissions could be quantified. This approach required knowledge of emission event timings to be known; something which is unlikely in real-world scenarios. The second approach assumed no prior knowledge of controlled release timings and is therefore likely to be a better reproduction of the real world, where exact knowledge of the timings of emission events is unlikely. For this second method, data were analysed in temporal intervals of varying lengths (e.g., daily [24-hourly], 12-hourly, 6-hourly etc.).

Inverse dispersion models are known to be sensitive to many user-defined parameters and inputs. The sensitivity of estimated emission rates to some parameters (e.g., release height, sensor height, $CH_4$ atmospheric background) was tested to assess the performance of the FEDS and the subsequent approaches to emission quantification. Further work on understanding the sensitivity of reverse dispersion models to multiple variables will be undertaken in the future.

## 3 Results and discussion

This section presents and discusses results from the assessment of the FEDS system.

### 3.1 Meteorology

Figure 3 shows hourly-averaged wind conditions (wind speed, wind direction, standard deviation in wind direction (1σ), and wind persistence) measured over the course of the assessment campaign.

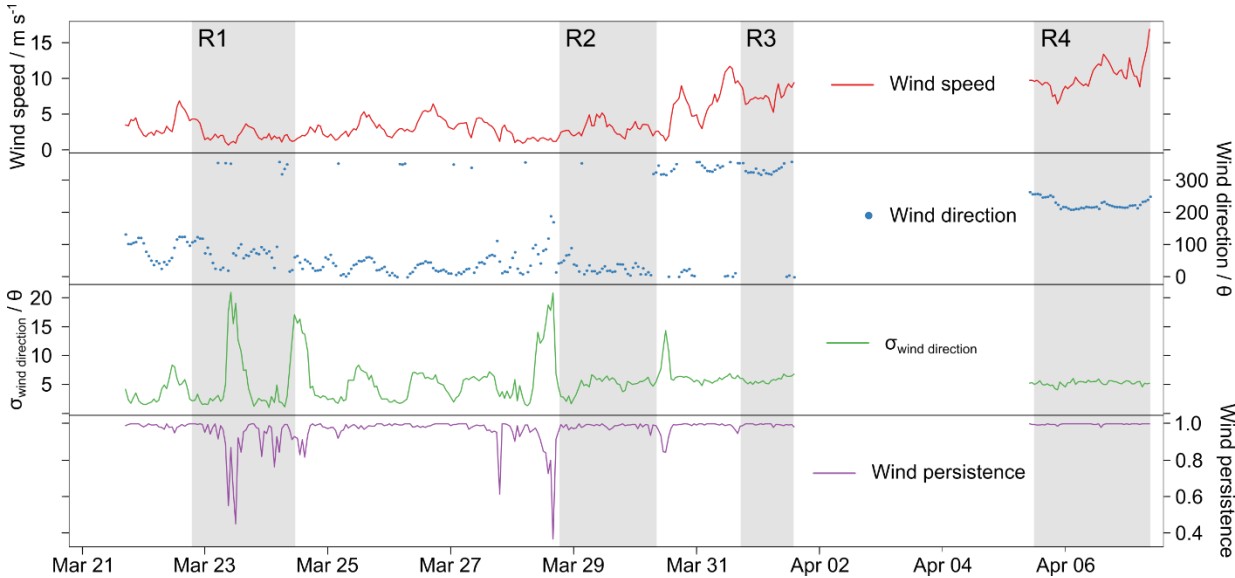

**Figure 3. Hourly-averaged wind speed, wind direction, standard deviation (1σ) in wind direction (σ_wind direction), and wind persistence (see Eq. 1) measured at 10 m above ground from the hydraulic mast on the FEDS trailer. Meteorology was not measured between 2nd and 5th April as the trailer was shut down. The four controlled release periods (R1 through R4) are shown in grey.**

Wind speed was generally below 5 m s$^{-1}$ before 30th March and above 5 m s$^{-1}$ after 30th March. The wind direction before 30th March was generally northeasterly, switching to northerly between 30th March and 2nd April, and then south-westerly on 5th and 6th April. Figure 4a shows a wind rose for the campaign. Periods of high standard deviation in wind direction occurred when wind direction was highly variable across the hour averaging period. Periods of lower wind persistence (e.g., <0.9) were typically associated with periods of higher standard deviation in wind direction, as expected. All occurrences of wind persistence below 0.9 were associated with average wind speeds of less than 2.5 m s$^{-1}$. Higher wind speeds (> 5 m s$^{-1}$) were almost always concurrent with wind persistence greater than 0.95. Mean hourly wind persistence (and range in wind persistence) across each of the four controlled releases is provided in Table 3.

### 3.2 Methane mole fraction results

Figure 4 shows a broad picture of CH$_4$ mole fractions measured from each sampling inlet location under differing wind directions. The CH$_4$ rose plots are arranged around the CRF release point location corresponding to the relative sampling

location (see Figure 2). The wind rose plot (also shown on Fig. 4) indicates the dominance of certain wind directions (northerly and northeasterly) during the campaign. Higher $CH_4$ mole fractions (>3 µmol mol$^{-1}$ ) were measured at inlets located to the south and southwest of the CRF (i.e., inlets 7, 9, and 12) corresponding to those locations downwind of the CRF during northerly and northeasterly winds. High $CH_4$ mole fractions were also measured at inlet 1 (located to the northeast of the CRF) during south-westerly winds, and at inlets 10 and 11 (located to the west of the CRF) during easterly winds. Those inlet

locations which were not downwind of the CRF generally measured lower $CH_4$ mole fractions (<2.5 µmol mol$^{-1}$) during northerly and northeasterly winds. These measurements demonstrate that the controlled release of $CH_4$ was correctly sampled by the FEDS as it cycled through sampling locations.

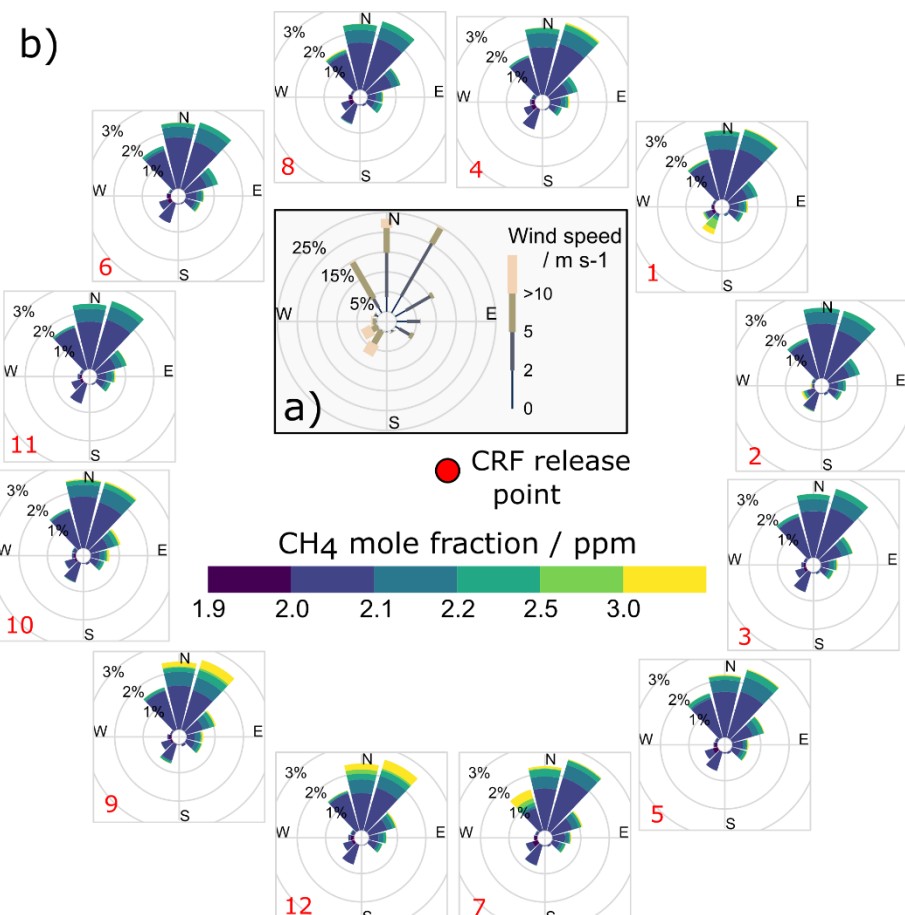

**Figure 4. a) Wind rose showing the frequency of wind speeds in 12 wind direction bins for the whole campaign period, and b) $CH_4$ rose plots showing $CH_4$ mole fractions measured at each inlet (arranged around the central CRF release point location as shown in Figure 2) during the campaign period. The radii of the wedges in 12 wind directions shows the frequency of mole fractions occurring in the coloured mole fraction bins. The red text indicates the inlet number corresponding to the $CH_4$ mole fraction rose.**

### 3.3 Quantifying emissions using prior knowledge of release timings

In this section, we assess the performance of the dispersion models for quantifying $CH_4$ emissions using prior knowledge of the timings of the controlled releases of $CH_4$. In this way, the data supplied to, or output from, the models was directly related to the periods of time in which $CH_4$ was being released. For Airviro, the model time step was set to cover the release period for each individual release. For WindTrax, the hours of output data corresponding to release periods were averaged. For the Gaussian plume analysis, only data within the release periods were considered.

Table 3 shows the mean controlled release rates from the four releases, and the emission rates quantified using the different dispersion models. The wind persistence (mean of hourly wind persistence values) for the duration of each controlled release is also provided.

**Table 3. Controlled (mean) $CH_4$ release rates, wind conditions, and quantified emission rates (Airviro[a], WindTrax[b], and Gaussian**
**plume[c]) for each of the four controlled releases. A value of NA indicates that no emission rate was quantified by that model. The quoted uncertainties for the wind persistence, wind speed, and controlled release rates are 1 standard deviation (1σ) about the mean across the release period. The model error output by Airviro is described as a standard error, defined as the standard deviation of the estimated error in Wickström et al. (2010). The model error output quoted here for WindTrax is the standard deviation (1σ) in individual hourly emission rates over each release period. The uncertainty value for the Gaussian plume model is the combined**
**uncertainty in the Gaussian plume equation (see Appendix B).**

| Release rate number | Approximate duration / hours | Wind persistence (range) | Wind speed / m s⁻¹ | Controlled release rate / kg h⁻¹ | Airviro [a] / kg h⁻¹ | WindTrax [b] / kg h⁻¹ | Gaussian plume [c] / kg h⁻¹ |
|---|---|---|---|---|---|---|---|
| R1 | 40 | $0.92 \pm 0.12$ (0.45 – 1.00) | $2.0 \pm 1.0$ | $0.991 \pm 0.011$ | NA | $1.96 \pm 6.20$ | $0.30 \pm 0.19$ |
| R2 | 36 | $0.99 \pm 0.01$ (0.95 – 1.00) | $3.1 \pm 1.1$ | $0.994 \pm 0.011$ | $0.79 \pm 0.12$ | $0.82 \pm 1.05$ | $0.69 \pm 0.36$ |
| R3 | 21 | $0.99 \pm 0.01$ (0.94 – 1.00) | $7.4 \pm 1.5$ | $0.993 \pm 0.011$ | $1.08 \pm 0.16$ | $1.72 \pm 1.89$ | $0.95 \pm 0.41$ |
| R4 | 43 | $0.996 \pm 0.004$ (0.98 – 1.00) | $10 \pm 1.9$ | $0.969 \pm 0.012$ | $1.08 \pm 0.09$ | $0.99 \pm 0.70$ | $0.91 \pm 0.38$ |

[a] Airviro model settings (see Appendix B): sampling lag time = 90 s; release height = 2 m; emission zone = 3 (see Appendix C); $CH_4$ background value = internally calculated.

[b] WindTrax model settings: sampling lag time = 90 s; release height = 2 m; point source emission; particle number = 50,000; $CH_4$ background value = 2.00 μmol mol⁻¹.

[c] Gaussian plume model settings: sampling lag time = 90s; wind direction bin width = 2°; $CH_4$ background = 5th percentile of data within release window (1.99 – 2.03 μmol mol⁻¹); median stability class.

All three models estimated emission rates within a factor of two of the controlled release rates, except during R1. Airviro was unable to output an emission rate for R1, whilst the Gaussian plume model estimated an emission rate of 0.30 ($\pm$ 0.19) kg h$^{-1}$.

WindTrax also performed poorly for R1, with a mean estimated emission rate of 1.96 ($\pm$ 6.20) kg h$^{-1}$. This may be explained by the wind persistence during R1, which was much lower (0.92) than for the other three release experiments (0.99, 0.99, 0.996 for R2, R3, and R4 respectively). The higher wind persistence (0.996) observed for R4 correlated with better performance from all three models, with estimates of 1.08 ($\pm$ 0.09), 0.99 ($\pm$ 0.70), and 0.91 ($\pm$ 0.38) kg h$^{-1}$ for Airviro, WindTrax, and the Gaussian plume simulation respectively.

Whilst the mean emission rates estimated by WindTrax were within a factor of two of the controlled release rates, the individual hourly emission rates varied considerably over each of the release periods. For example, hourly estimates of emission rates for R1 ranged between 0.006 and 33.4 kg CH$_4$ h$^{-1}$, thus explaining the large standard deviation about the mean for R1. If shorter controlled releases (on the order of a few hours) were used, any results may therefore have been subject to large absolute biases. However, it should be noted that the WindTrax setup used in this study was not optimal; WindTrax

typically requires parity between the number of emission sources and the number of concentration measurement locations (i.e., one concentration measurement for each emission source). The design of the FEDS assessment campaign meant that this was not possible and therefore WindTrax was subject to increased model uncertainty. Furthermore, a default value for atmospheric turbulence was used for all WindTrax simulations. Catering WindTrax model simulations to conditions at the time of each release, for example by using different Pasquill stability classes, may have improved results. A single, constant value for

turbulence may also have precluded the ability for WindTrax to model diurnal variations in atmospheric stability. The use of measured temperature data in Airviro may have aided that model's treatment of atmospheric turbulence and stability.

The source location (SL) module in Airviro (see Appendix B) also has the capability to estimate the likely emission source location(s) based on the input data. Figure 5 shows the probability of emission source locations output by Airviro. It is clear that for R1, Airvrio was incapable of correctly locating the emission source location. This is potentially also correlated

with the lower wind persistence during R1 (0.92 $\pm$ 0.12), and likely explains the lack of an output emission rate for that release (Table 3). For R2, R3, and R4, Airviro was able to estimate the location of the emission source, albeit with greater confidence during R4 than R3, and during R3 than during R2. The increased confidence with locating the emission source correlated with higher wind persistence, as well as with reduced percentage uncertainty in the emission rate. This implies that Airviro performs better under more consistent wind conditions, and that emission source locations and quantified emission rates under

inconsistent wind conditions may be subject to greater uncertainty.

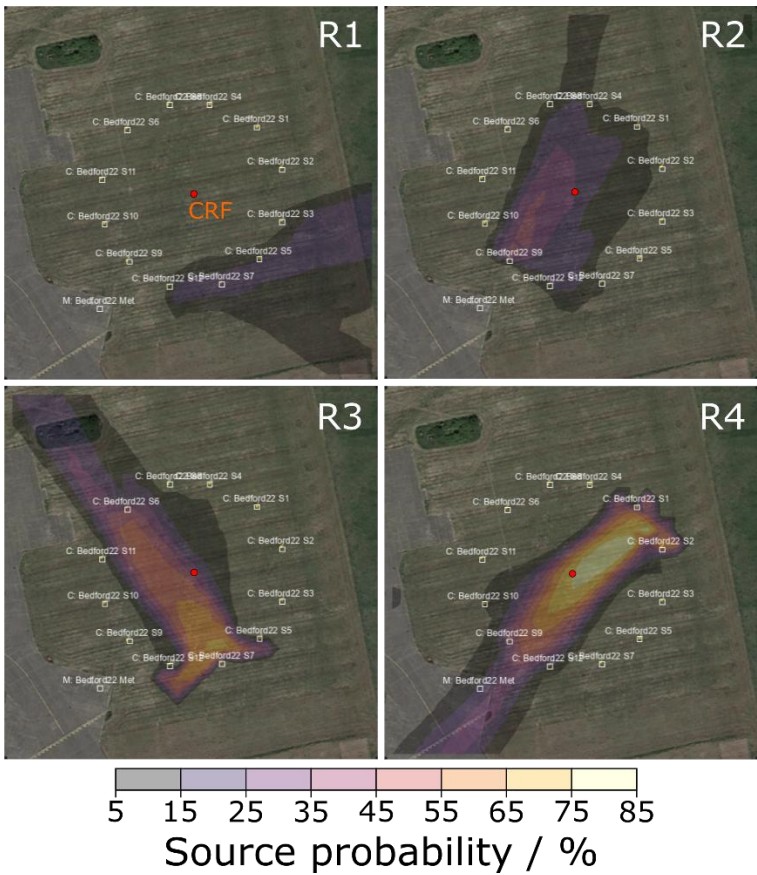

**Figure 5. Airviro output 'source probability' for the four release periods (R1, R2, R3, and R4). The coloured contours show the probability that the emission source is located within each contour shown by the colour scale. The red dot shows the true location of the controlled release facility (CRF) from which CH$_4$ was released. Satellite imagery: Google, ©2020 Maxar Technologies.**

Figure 6 shows the Gaussian plume functions for each of the four controlled releases, following binning of the CH$_4$ data into 2° wind direction bins. The correlation between CH$_4$ enhancements and the wind direction blowing directly towards the sampling inlets (i.e., rotated wind direction closer to zero) was clear in the case of R2, R3, and R4. However, there was little correlation between CH$_4$ enhancement and wind direction for R1. Despite this, a Gaussian curve was still generated, and the method estimated a final emission of $0.30 \pm 0.19$ kg h$^{-1}$. Emission rates were estimated using the highest point of the Gaussian curve (a1; see Appendix B). Whilst the result for R1 was lower than the controlled release rate, it was still within an order of magnitude, though this may be a coincidence rather than proof that this method works even under poor wind conditions. The spread of CH$_4$ enhancement data for the other three releases was much tighter, clearly demonstrating that enhancements were only measured at sampling locations when the wind was blowing towards them (see Fig. 4). As was the case for the other two dispersion models, the Gaussian plume estimated emission rate and the calculated uncertainty appeared to be correlated with the wind persistence for the duration of each controlled release.

It should also be noted that averaging concentration data into wind direction bins over long periods of time may combine $CH_4$ enhancements measured under very different wind speed conditions. This may explain the high variability in $CH_4$ data measured even within the plotted Gaussian function (see grey data in Fig. 6). This simple Gaussian plume approach may therefore be further refined by ensuring that the wind field, including the wind speed, is consistent for the duration of any analysed time periods.

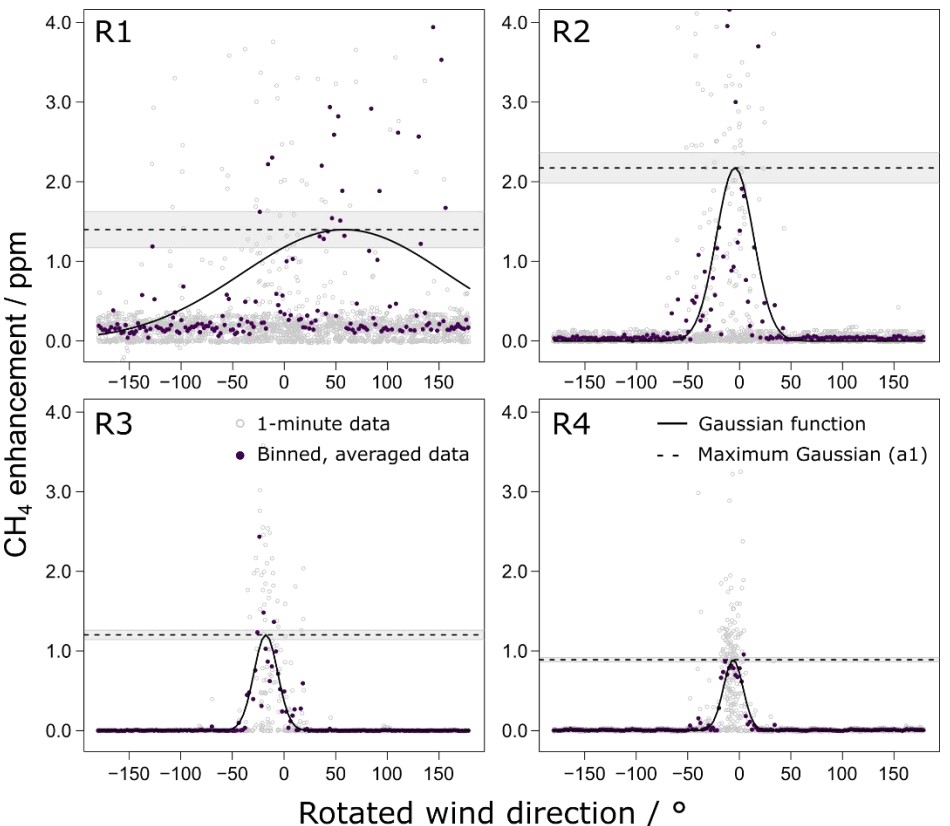

**Figure 6. OTM-33A analysis and fitted Gaussian plume function for the four controlled releases, using a bin width of 2° and a $CH_4$ background value equal to the 5th percentile of $CH_4$ data during each release. The maximum $CH_4$ enhancement was determined by the maximum point of the Gaussian function (see Appendix B), and is denoted here by the dashed black line, with the uncertainty in this value shown in grey.**

### 3.4 Quantifying emissions without prior knowledge of release timings

In this section, the performance of the reverse dispersion models for quantifying emissions without prior knowledge of the release timings is assessed. This approach more closely mimics a real-world scenario, in which precise (or even approximate) knowledge of emission event timings may be unknown. The impact of changing different model variables and parameters on model performance and sensitivity is also examined in this section. Only Airviro and WindTrax are analysed, as the custom

Gaussian plume model (based on OTM-33) has less scope for varying parameters. Some additional model parameters are also

analysed in Appendix C but will be the focus of future work assessing the sensitivity of dispersion models to multiple variables.

### 3.4.1 Changing model time step and averaging period

In the absence of information regarding the timings of emission events, it is expected (and plausible) that dispersion models will be used to evaluate data over regular periods of time. Airviro has scope for setting the time period (via a start and end time for a simulation), corresponding to multiple time intervals of input data, over which data is evaluated. WindTrax only outputs

emission rates at intervals equal to the timestep of input data (in this case, hourly). Whilst this does not allow for equivalent comparison between the two models, the 1-hourly data input to WindTrax was averaged over longer periods of time (e.g., daily) to allow for indirect comparison to the changing Airviro timestep length. Time averaging for WindTrax was therefore applied to both the input data (concentration and meteorology) and to the output data (emission rate).

Figure 7 shows time series of Airviro and WindTrax emission rate estimates (where emission rates were calculated

either by time-averaging hourly input data prior to model evaluation, or by time-averaging hourly output data following model evaluation) compared with CRF release rates, for 3-hourly and 12-hourly periods (see Appendix C for time series of other averaging periods). Whilst the average emission rate estimated by the models for the duration of a controlled release was broadly similar with the release rate (Table 3), there was a lot of variability in the emission rates estimated by both models over shorter time periods. This variability was far more evident for shorter averaging periods (i.e., in the 3-hour average

emission rates compared with the 12-hour emission rates). For example, WindTrax estimated emission rates ranged between 0.003 and 33.4 kg h$^{-1}$ on an hourly basis. The impact of wind persistence is also visible on these time series, especially for Airviro. Airviro failed to estimate any emissions during the first controlled release (R1; wind persistence 0.92).

The capacity of the models for correctly capturing emission events (and periods without emissions) can be evaluated using contingency metrics (Bennett et al., 2013). The righthand panels in Figure 7 show the percentage of true positives (in

which the model correctly identifies emissions), false positives (in which the model incorrectly identifies emissions), true negatives (in which the model correctly identifies periods of no emissions), and false negatives (in which the model incorrectly identifies periods of no emissions) for both Airviro and WindTrax using different timesteps. For this analysis, an arbitrary threshold of 0.05 kg h$^{-1}$ was chosen, above which output emission rates were considered an event, and below which output emission rates were considered a non-event. A threshold was selected to exclude negligible emission rates (e.g., 0.003 kg h$^{-1}$)

from counting as an emission event detection. Airviro showed a large proportion of false negatives in all time averaging cases; false negatives were generally associated with R1 and therefore potentially related to the lower wind persistence during that release. WindTrax had fewer false negatives (emissions were correctly identified during R1 even if the magnitude wasn't well constrained) but a greater proportion of false positives, where emissions above the arbitrary threshold (>0.05 kg h$^{-1}$) were estimated when no controlled releases were taking place. It should be noted that the lack of meteorological data between R3

and R4 meant that model contingency during this period was not evaluated. Airviro had a greater percentage of correct results (true positives and negatives) for 12-hourly and 6-hourly timesteps, whereas WindTrax had the greatest proportion of correct

results when using daily averaging, and the lowest proportion of correct results when using 6-hourly averaging. It should be noted that the relative proportion of time spent with and without controlled releases of emissions may bias the results in favour of one or other of the models; a model which never output an emission rate would still yield true negatives some of the time whilst a model which always modelled emission rates greater than zero would give true positives. The time period when the FEDS trailer was shut down (2nd-5th April) was not analysed here (see Fig. 3).

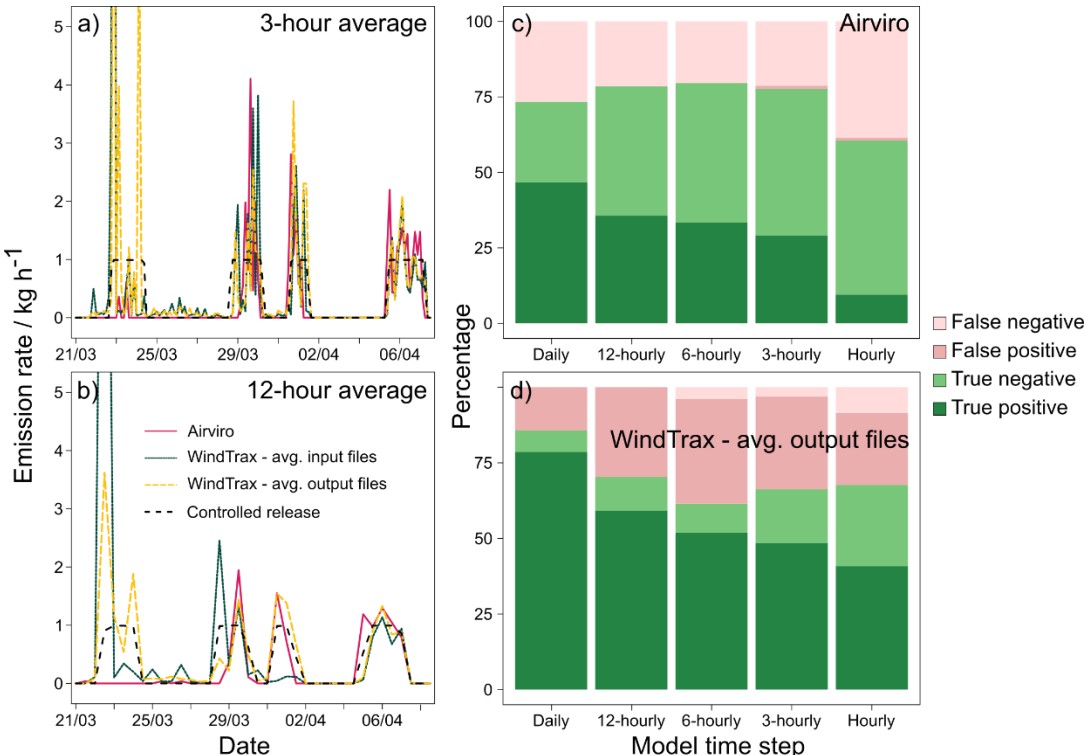

**Figure 7. a) Time series of 3-hour average emission rate results for Airviro and WindTrax (averaging either the input data or 1-hourly output results for WindTrax). The black dashed line shows 3-hour average controlled release rates. b) Time series of 12-hour average emission rate results for Airviro and WindTrax (averaging either the input data or 1-hourly output results for WindTrax). The black dashed line shows 12-hour average controlled release rates. c) Contingency matrix (event observation) for Airviro with different time steps. d) Contingency matrix (event observation) for WindTrax with different averaging of hourly output emission rates. All results shown with a lag time of 60 seconds and a plume height of 2 m.**

The success index provides a measure of the ability of a model to correctly detect occurrences and non-occurrences of emissions (Bennett et al., 2013). Table 4 gives the success index for Airviro and WindTrax with different timesteps or averaging periods. Airviro with daily, 12-hourly, and 6-hourly timesteps performed the best, despite being unable to detect emissions during R1. However, the success of Airviro dropped off with shorter timesteps. WindTrax performed most successfully using daily and hourly averaging but did show such a substantial reduction in success rate when using intermediate length averaging periods. Airviro had broadly greater success rates than WindTrax despite not estimating emission rates during

R1. This was because WindTrax output emission rates (and sometimes substantial emission rates) for all hours of input data regardless of if emissions were truly present or not (i.e., many false positives).

Whilst the success index provides an indication of correct interpretation of event timings, the success index does not provide information on the performance of the models for accurately estimating the rates of emission. Table 4 also provides a bias score (or mean error), and root mean square error (RMSE) for each of the model iterations. As controlled release rates were ~1 kg h$^{-1}$, the absolute values given in Table 4 can be easily translated to relative (%) errors. Airviro underestimated emissions on average (negative bias), though this may be exaggerated by the lack of estimated emissions during R1. WindTrax, on the other hand, overestimated emissions on average (positive bias). The RMSE was generally lower for Airviro than for WindTrax, largely due to the vast overestimation of some emission rates during R1 by WindTrax (especially when using shorter time averaging). WindTrax, with a daily averaging period, performed the best overall, with the lowest RMSE (0.33 kg h$^{-1}$; 33%) and an average bias of -0.02 kg h$^{-1}$ (-2.0%). Airviro with 12-hourly or 6-hourly periods performed next best, with a small negative bias (of -0.15 and -0.11 kg h$^{-1}$ (-15% and -11%) for 12-hourly and 6-hourly respectively), and RMSE scores of 0.51 and 0.53 kg h$^{-1}$ (51% and 53%) for 12-hourly and 6-hourly respectively. A timestep (or averaging period) of 1-hour performed the worst when compared with other timesteps for both models.

Table 4. Metrics comparing the Airviro and WindTrax modelled emissions rates with the controlled release emission rates whilst varying the time step (or averaging period).

| Model | Time step (or averaging period) | Success index / % | Bias / kg h$^{-1}$ | RMSE / kg h$^{-1}$ |
|---|---|---|---|---|
| Airviro | Daily | 82 | -0.30 | 0.54 |
| | 12-hourly | 81 | -0.15 | 0.51 |
| | 6-hourly | 81 | -0.11 | 0.53 |
| | 3-hourly | 78 | -0.12 | 0.63 |
| | Hourly | 59 | -0.32 | 0.75 |
| WindTrax | Daily | 67 | -0.02 | 0.33 |
| | 12-hourly | 64 | 0.14 | 0.62 |
| | 6-hourly | 57 | 0.18 | 1.06 |
| | 3-hourly | 65 | 0.17 | 1.38 |
| | Hourly | 68 | 0.19 | 2.39 |

**3.4.2 Wind persistence**

The impact of low wind persistence (or high wind direction variability) on emission estimation was observed during R1 in Section 3.3. Airviro, in particular, struggled to estimate emissions during those releases with lower average hourly wind persistence. There was a small correlation between the error in estimated emissions and the average hourly wind persistence

($R^2 = 0.58$; see Appendix C), with generally greater error in estimated emission rate during periods of lower wind persistence. Several thresholds for excluding estimated emission data were trialled, using values of average hourly wind persistence below which emission data for that time period were considered invalid. Results (success index, mean bias, and RMSE) following application of these thresholds are shown in Figure 8, for wind persistence threshold values of: no threshold (equivalent to all data), 0.85, 0.90, and 0.95. The proportion of time periods eliminated by the threshold increased with increasing threshold value.

Using a wind persistence threshold improved the success index for Airviro in all cases but reduced the success index for WindTrax in some cases (see Appendix C for contingency matrices before and after application of a wind persistence threshold). Improvement in success index for Airviro but not for WindTrax was likely due to the different performance of these models during R1. The first release (R1) occurred during a period of low wind persistence and was therefore invalidated when using a wind persistence threshold. As Airviro failed to output an emission rate during R1, removing R1 from the analysis improved the success index for the remaining releases.

Figure 8 clearly demonstrates the negative bias in emission estimates output by Airviro. The bias in Airviro results generally reduced when using any wind persistence threshold (especially with 12-hourly time steps). The positive bias in emission estimates is also evident for WindTrax. For WindTrax, a greater bias was observed when using wind persistence thresholds for 12-hourly and 6-hourly averaging, but smaller biases were observed when using wind persistence thresholds for other time averaging periods. The impact of including wind persistence thresholds on the RMSE was mixed for both models. For Airviro, reductions in RMSE occurred for daily time steps but there was little impact when using shorter time steps. For WindTrax, the RMSE reduced slightly when using a wind persistence threshold of 0.95 with daily time averaging but actually increased when using a wind persistence threshold of 0.95 with 12-hourly averaging. Interestingly, for WindTrax, the RMSE improved when using a wind persistence threshold with the shorter time-averaging periods. Appendix C provides values for success index, bias, and RMSE with wind persistence thresholds of 0.90 and 0.95.

Overall, daily averaging periods appeared to work best for both Airviro and WindTrax, with generally greater success index, and lower values for both bias and RMSE. Slightly shorter time periods (i.e., 12-hourly) appeared to be the next best averaging period. Averaging data over 12-hourly periods also had the benefit of providing a greater number of observations for analysis (28 12-hourly data points compared to 15 daily data points). The results whilst testing different wind persistence thresholds were more mixed between Airviro and WindTrax but a threshold of 0.95 had the lowest RMSE for both models under both daily and 12-hourly averaging. For these reasons, all experimental results following are demonstrated for 12-hourly averaging with a wind persistence threshold of 0.95.

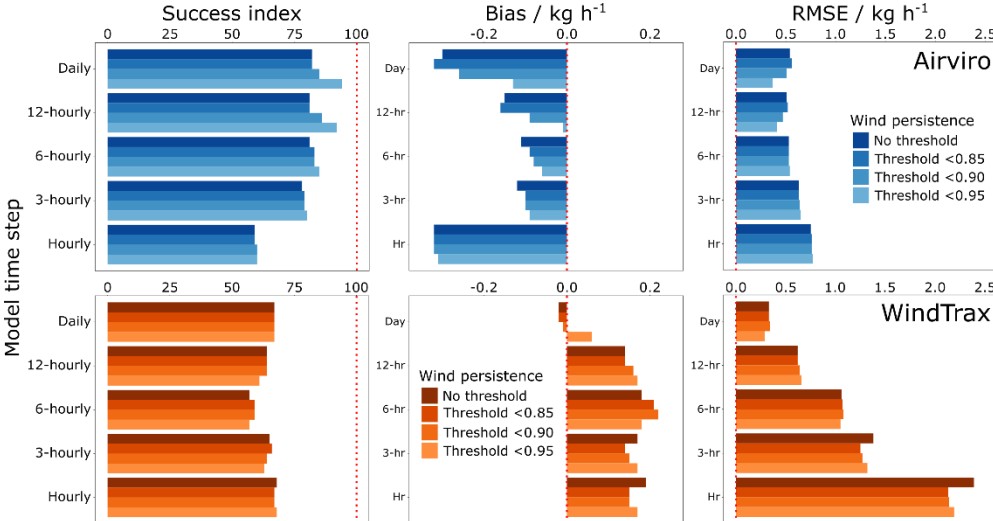

Figure 8. Graphical summary of success index (left column), bias (central column), and RMSE (right column) for Airviro (top row; blue) and WindTrax (bottom row; orange). Ideal values are indicated by red dashed lines (success index = 100%; bias = 0 kg h$^{-1}$; RMSE = 0 kg h$^{-1}$). Different shaded colours represent results with different average hourly wind persistence thresholds, where data was removed if the wind persistence was below a certain value.

### 3.4.3 Changing number of sampling locations

The number of sampling locations required to accurately monitor an emission source is a key parameter for the FEDS system and any other system with a distributed sampling network. The more locations installed, the more equipment and resources required and the longer the time taken to deploy that equipment. A minimum number of sampling locations would be preferential in most cases to reduce equipment costs and labour.

Figure 9 shows the RMSE (kg h$^{-1}$) results for Airviro and WindTrax simulations with data input from differing numbers of sampling locations. A clear trend was observed with RMSE decreasing in magnitude with increasing numbers of sampling locations. A similar trend was observed for the success index (data point colours), with model simulations more successful at correctly identifying emission events when data from more sampling locations were used. The exact configuration of locations, along with RMSE and success index scores, can be found in Appendix C.

Interestingly, the RMSE was lowest for WindTrax when using data from three sampling locations. Whilst the success index was also low (and therefore quite poor) for these simulations, the low RMSE values may be the result of the WindTrax model's optimisation procedures. WindTrax seeks to resolve a least-squares statistical fit to the data based on the geometric arrangement of sensors and sources; having more emission sources than concentration sensors can lead to an overdetermined system of equations and thus greater uncertainty (Crenna et al., 2008).

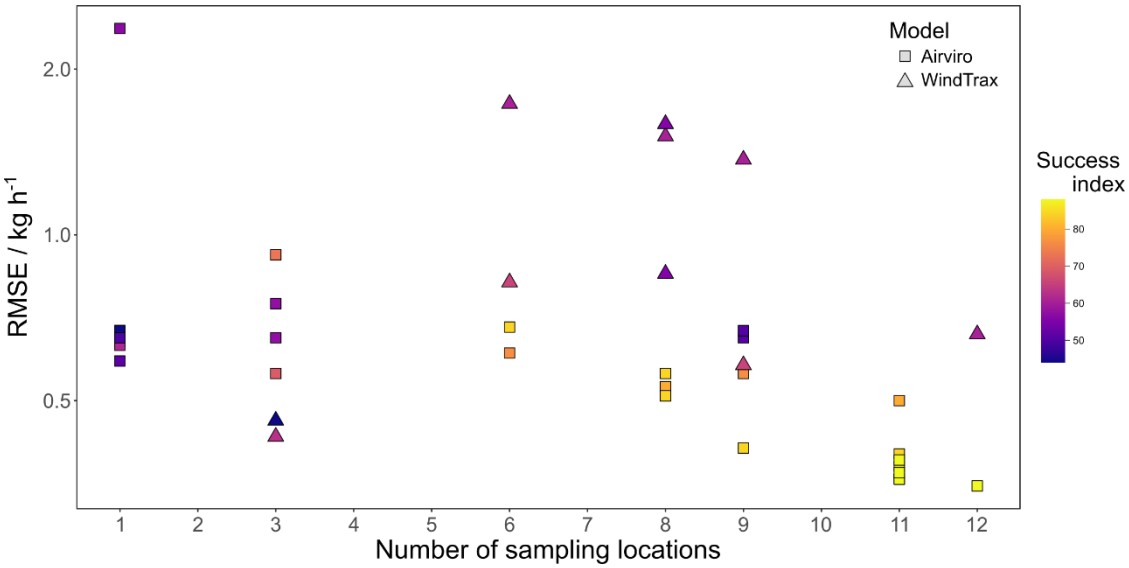


**Figure 9. RMSE results as a function of the number of sampling locations for both Airviro (square data points) and WindTrax (triangular data points). Data points are coloured by the success index.**

### 3.5 Comparison with other continuous emission monitoring solutions

Other work has sought to assess the ability for continuous emission monitoring solutions to detect and quantify $CH_4$ emissions using controlled releases. In this section, the FEDS system is compared against reported metrics for other analogous continuous monitoring solutions i.e., those which use sampling networks (and therefore not including systems making use of scanning or imaging). It should be noted that the literature available for such reported metrics often used many more controlled releases, and across a greater range of release rates, than those used in this work.

Bell et al. (2023) analysed 11 continuous monitoring solutions, of which six used point sampling, and of which five of those six provided emission quantification estimates. Several hundred controlled releases were performed at the Methane Emissions Technology Evaluation Center (METEC), a Colorado State University facility containing equipment typically found at natural gas production facilities. Leaks at the METEC facility are simulated at connectors and flanges using regulated flows to mimic real-world emission scenarios (Ravikumar et al., 2019). Release rates within the Bell et al. (2023) study were between

0.0004 and 6.4 kg h⁻¹ and were blinded to participants. Details of the continuous monitoring solutions assessed were sparse to preserve anonymity and hence direct methodology comparisons with the FEDS system are difficult. True positive detections for these solutions ranged from 0.3% to 87.7% indicating a wide range of abilities to detect emissions. However, some of the solutions with the highest true positive detections also had large false positive detections (the solution with 87.7% true positive had 79.1% false positive detection) indicating that some solutions identified emissions when non were present. For those

solutions which quantified emissions, the mean relative quantification error ranged from -39.5% to 50.5%, with the most accurate solution having a mean relative error of 10.2%.

Ilonze et al. (2024) performed a follow-up to the single-blind controlled release testing in Bell et al. (2023), testing multiple continuous monitoring solutions over 11 weeks of trials. True positive rates reported in Ilonze et al. (2024) ranged from 29% to 73%. Wind speed was noted to significantly influence the probability of detection for 5 out of 9 solutions. For

controlled release rates >1 kg $CH_4$ $h^{-1}$, single estimates ranged from 0.08 to 18 times the true emission rate. The mean relative quantification error ranged from -12.8% to 256%, with the most accurate solution having a mean relative error of -12.8%. Quantification of facility-scale emissions (simulated through multiple release points) were more promising, with most solutions within a factor of two of the total release rate. Across the metrics evaluated, most of the monitoring solutions assessed in both Bell et al. (2023) and Ilonze et al. (2024) improved.

Day et al. (2024) performed controlled releases at both a dedicated facility (METEC) and at real oil and gas facilities. Seven continuous monitoring solutions using point sampling were tested, with four reporting emission rates. No solution achieved 90% probability of detection and accuracy of estimated emission rates were low with mean relative errors of between -25% and -70% across the four solutions. There was a significant difference in performance between results from METEC (Ilonze et al., 2024) and the results at real facilities suggesting that controlled releases at test facilities may not adequately

simulate real-world scenarios.

Mbua et al. (2025) performed multiple releases of $CH_4$ with release rates between 0.01 and 8.7 kg $h^{-1}$ and durations between 10 s and 8 h. $CH_4$ was measured using a single ABB LGR Microportable Greenhouse Gas Analyzer (MGGA) sampling at 10 Hz. Two reverse dispersion models were tested: one using a Gaussian plume inversion and one using a Lagrangian stochastic approach (WindTrax). The Gaussian plume model struggled to quantify emissions when multiple

releases were run simultaneously but performed well for single controlled releases. Gaussian plume models were also reported to perform better when measurements were taken at greater distances downwind of the emission source (and the plume was therefore well mixed) (Riddick et al., 2022b). Quantification accuracy using WindTrax was higher than for the Gaussian plume model for single releases but reportedly low for multiple simultaneous releases (Mbua et al., 2025). Mbua et al. (2025) noted challenges in performing controlled releases, with accumulation of residual $CH_4$ between releases leading to interference. This

highlights the complexity of performing such experiments.

Testing of several commercial monitoring solutions in Europe recently took place at the TotalEnergies Anomalies Detection Initiatives (TADI) facility (McManemin et al., 2025). TADI is located on an industrial site with dismantled oil and gas equipment and is capable of simulating emissions with release rates between 0.004 and 1080 kg $CH_4$ $h^{-1}$. Of the eight commercial solutions tested, only one solution provided continuous monitoring from multiple ($n = 12$) fixed-point sampling

locations. This solution achieved a true positive detection rate of 91% and false positive rate of 1% across 147 releases. However, whilst the solution performed well in terms of detection, it significantly underestimated the true value of emission rates especially for releases >100 kg $h^{-1}$ (linear regression gradient = 0.14; McManemin et al., 2025).

Where other work evaluating the efficacy of continuous emission monitoring solutions have focussed on the method as a whole, this work looked at the impact of applying different reverse dispersion models for emission detection and quantification to the same set of measurement data. The probability of detection (a metric indicating successful detection of controlled releases) was 75% for Airviro (three out of four releases identified), 100% for WindTrax (four out of four identified), and 100% for the simple Gaussian plume solution (four out of four identified).

The performance of the three independent reverse dispersion models for the accurate estimation of emission rates was tested in this work. Individual hourly emission estimates from all models had high error (high RMSE) when compared to controlled release rates, but estimates were greatly improved when data was averaged over time periods greater than 12-hours (lower RMSE). The performance of all three models was also shown to be closely correlated to the wind conditions, with a highly variable wind field (assessed via wind persistence) generating circumstances in which the models performed less well, or with greater uncertainty. Higher wind persistence (or low variability in wind direction) during release periods improved performance. This suggests that the wind direction variability must be low for the course of a full measurement cycle (measuring from all sampling locations) and is likely an important factor in any continuous monitoring solution.

The performance of the models was also improved when using prior knowledge of the timings of emission events. However, it is acknowledged that this information may be unknown in real-world scenarios. This work provides evidence that Airviro and WindTrax generally worked well in the absence of knowledge of the timings of releases; emission rates were quantified reasonably accurately (low bias, low RMSE) using daily and 12-hourly averaging of input data ($CH_4$ concentration and meteorology), especially for Airviro, and especially after removing periods of lower wind persistence below a threshold of 0.95. For daily averaging periods (and for a controlled release rate of 1 kg h$^{-1}$), the RMSE was 0.37 kg h$^{-1}$ (37% relative error) and 0.29 kg h$^{-1}$ (29%) for Airviro and WindTrax respectively whereas when using hourly averaging periods, the RMSE was 0.77 kg h$^{-1}$ (77%) and 2.19 kg h$^{-1}$ (217%) for Airviro and WindTrax respectively. The mean relative biases and RMSE values reported here are broadly similar to those reported by Bell et al. (2023) and Ilonze et al. (2024). However, the derived statistics from this and similar studies may not be directly comparable. For example, the biases and RMSE values calculated in this work were calculated from multiple emission rate estimates (using different time-averaging periods) across (and within) four distinct emission events (i.e., one release = multiple estimates). This differs from the biases and RMSE values reported in Bell et al. (2023) and Ilonze et al. (2024), which were calculated from single emission rate estimates for each of many (>100) individual emission events (i.e., one release = one estimate).

Of the three reverse dispersion models tested, Airviro (a Gaussian dispersion model) performed the best and most consistently, as well as offering additional advantages in being able to provide estimates of the emission source location. WindTrax (a Lagrangian stochastic model), on the other hand, required the emission source location to be specified before emissions could be quantified. This contrasts with Mbua et al. (2025), where WindTrax was found to perform better than the Gaussian plume used in that work.

There were substantial limitations to the controlled release experiments performed in this work. Controlled releases were relatively simple, with only one release rate tested from a single release height (2 m above ground). Release durations

were quite lengthy (>24 h) which may not reflect real-world emission scenarios where emissions are typically sporadic and temporally varied. The relatively short distance between measurement location and controlled release in this work (~38 m) may also have negatively impacted the accuracy of the dispersion models (particularly Airviro, a Gaussian model which relies on the measurement of well-mixed emission plumes). Alternative models capable of modelling transport on small spatial scales (<1 km) exist, and FEDS could be used to validate these models for estimating $CH_4$ emissions. Further work is also required to test the sensitivity of these dispersion models (and output emission rates) to user-specified input parameters, such as the $CH_4$ background concentration or particle number. Some work to this effect is covered in Appendix C but is not covered here in great detail.

**4 Conclusions and future improvements**

NPL's Fugitive Emission Distributed Sampling (FEDS) system is a mobile, multi-inlet sampling system. FEDS has important advantages over other network systems in that only a single high-performance trace gas analyser is required, reducing the cost relative to systems using multiple high-performance instrumentation. Controlled releases of $CH_4$ were used to assess the FEDS system for measuring and estimating $CH_4$ emission rates from point sources on the facility scale. Data were collected from each individual sampling location for only a few minutes each hour, but hourly-averaged data was suitable for input to reverse dispersion models for emission source location and emission rate estimation.

The use of only a single trace gas analyser did come at the cost of reduced temporal coverage at each sampling location. Only four minutes of data were collected at each sampling location every hour (with at least the first 60 seconds of data discarded after switching inlets in case of contamination in the optical cell due to slow cell flushing time). This remains the major disadvantage of using only a single analyser measuring from multiple locations over using multiple analysers in a FEDS-type system. However, it should be noted that the use of multiple analysers may introduce an element of inter-instrument variability to the data. Such variability may be mitigated with robust calibration procedures but may not be ruled out, especially if analysers are of different types and/or manufacturers. In the absence of a direct comparison between using one and using multiple analysers, it is difficult to conclusively determine if measurement quality is proportionately improved when using a more expensive system with multiple analysers.

Future FEDS development will trial the incorporation of low-cost sensors for the continuous measurement of $CH_4$ at each of the sampling locations. This will increase the temporal capture of $CH_4$ concentration at each location up to 100%, albeit with a potentially reduced data quality (relative to the performance of the analyser). The four minutes of co-located data measured by the gas analyser could be used to correct the low-cost sensor data to improve data quality. However, given current concerns around low-cost sensor reliability for accurate measurement of $CH_4$, it is not expected that low-cost sensors could be used as a direct substitute for a high-performance gas analyser at this point in time. Further improvements to the FEDS system may be realised via the use of gas analysers with increased cell flushing times (such as the >10 Hz analysers often used for eddy covariance measurements) to reduce the sampling lag time when switching between sampling locations. Additionally, collecting more wind data (e.g., by including a wind anemometer at each sampling location) may substantially improve the modelling of the wind field, which is key for accurate emission estimation and source localisation (particularly in complex environments).

The assessment approach used a limited set of controlled release experiments of $CH_4$ which did not replicate the range of parameters present for real-world emission sources. Controlled release experiments should be greatly expanded to fully validate the FEDS system for real-world emission scenarios from simulated oil and gas facilities or landfill-type (area) emission sources. For example, controlled releases in this work were continuous (same release rate) over reasonably long periods of time (>20 hours) and hence the performance of FEDS for transient or variable emission events of considerably shorter duration (i.e., hours or minutes) was not assessed. However, we provide data and evidence for the quality assurance of quantified

emissions output by local scale reverse dispersion models, largely through interpretation of the wind field. Future assessment

experiments could also seek to examine the impact of other variables on emission quantification performance including the release rate of emissions (continuous or variable), release duration, sensor or receptor height(s), topography, and obstructions to dispersion.

  **Appendices**

**Appendix A: Additional contextual information**

Table A1 provides the manufacturer-reported specifications of the high-performance gas analyser used in this work (specifications were taken from LGR's '*Quickly Locate Natural Gas Leaks Anywhere: Fast Methane/Ethane Analyzer ($CH_4$, $C_2H_6$)* datasheet provided in internal communication).


**Table A1. Manufacturer-reported specifications of the LGR-FMEA (GLA331 series). Note that manufacturer-reported specifications likely apply to laboratory conditions and not to conditions during field trials.**

| | |
|---|---|
| Manufacturer | ABB Ltd. |
| Analyser type | Fast Methane-Ethane Analyzer (GLA331 series, Model 909-0027) |
| Operating principle | Off axis-integrated cavity output spectroscopy |
| Gases measured | Methane ($CH_4$) |
| | Ethane ($C_2H_6$) |
| Methane precision | 2 ppb (1σ, 1 sec); 0.6 ppb (1σ, 10 sec) |
| Methane accuracy | <1% [of reading] without calibration (between 10 ppb and 100 μmol mol$^{-1}$ $CH_4$) |
| Methane resolution | 0.01 ppb |
| Methane drift | Not specified |
| Methane range | Up to 100,000 μmol mol$^{-1}$ |
| Measurement rate | 1 Hz |
| Weight | 29 kg |
| Dimensions | 40 × 48 × 61 cm |

The specifications of the LGR FMEA may be compared with lower cost sensors such as the Cubic SJH-5A Methane Sensor (which uses non-dispersive infrared (NDIR) for $CH_4$ detection), or the Gazomat CATEX™ 3 (a catalytic combustion sensor). Such low-cost sensors may be preferred for some applications but at the expense of performance and quality of data. The Cubic SJH-5A has a (manufacturer-reported) $CH_4$ measurement range of 0%-5% (by volume), accuracy of ±0.06% in the range 0%-1% (by volume), measurement resolution of 0.1%, and response time of <25 seconds. The measurement resolution is particularly poor when compared with the LGR FMEA. The specifications of the Cubic SJH-5A sensor mean that detection and identification of all but the largest emission events might be challenging. The CATEX™ 3 has a reported lower detection limit for $CH_4$ of 100 μmol mol$^{-1}$ and accuracy of ±100 μmol mol$^{-1}$ in the range 0-1000 μmol mol$^{-1}$. The detection limit and accuracy are much lower than that for the LGR FMEA and may limit application of this sensor for detecting small emissions.

These two sensors were examined here for illustrative purposes only, to demonstrate the higher performance of higher cost gas analysers and the improved data quality.


Table A2 shows the proportion of time spent in each Pasquill stability class during each of the four controlled releases. The Pasquill stability class was calculated using the method prescribed in OTM-33A (US EPA, 2014) from the measured standard deviation in wind direction.

**Table A2. Proportion of time (1-minute average) spent in each Pasquill stability class for each of the four controlled releases.**

| Release rate number | Stability class / % | | | | | | |
|---|---|---|---|---|---|---|---|
| | **1** *Extremely unstable* | **2** *Moderately unstable* | **3** *Slightly unstable* | **4** *Neutral* | **5** *Slightly stable* | **6** *Moderately stable* | **7** *Extremely stable* |
| R1 | 1.7 | 0.7 | 1.4 | 1.9 | 3.8 | 8.6 | 81.8 |
| R2 | 0.0 | 0.0 | 0.0 | 0.0 | 0.4 | 8.6 | 91.0 |
| R3 | 0.0 | 0.0 | 0.0 | 0.0 | 0.2 | 8.1 | 91.7 |
| R4 | 0.0 | 0.0 | 0.0 | 0.0 | 0.1 | 5.6 | 94.3 |

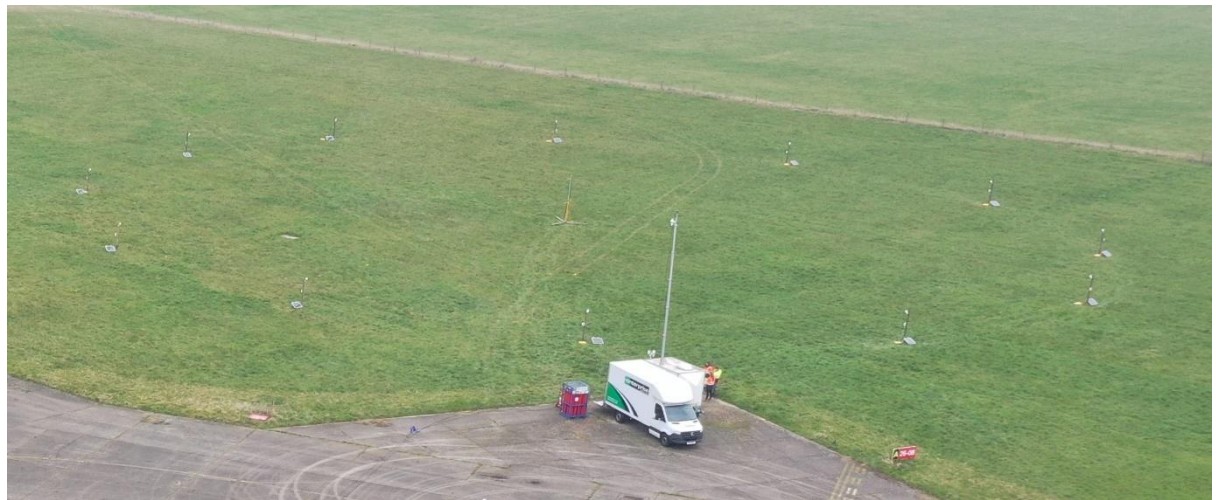

**Figure A1. Aerial photograph of the field setup (see also Fig. 2). The FEDS trailer can be seen in the centre foreground of the image**
**(behind the van) with the 10 m meteorological mast attached to the trailer. The 12 sampling locations are arrayed in a circle on the grass beyond the trailer with the CRF release location in the centre.**

Figure A2 shows a simple illustration of periods during which $CH_4$ concentration was elevated above the local background ($CH_4$ >2.5 µmol mol$^{-1}$) with data plotted as back trajectories from the sampling location using the concurrent wind direction. Data is overlaid onto a graphical representation of the FEDS setup, with the CRF release location shown in the centre in red (see Fig. 2). It is clear that the majority of plotted trajectories passed close by to the CRF release location, indicating that enhancements of $CH_4$ were typically observed downwind of the release point. However, multiple instances of elevated $CH_4$

did occur during wind directions which had not passed close to the CRF release point. Many of these instances were correlated with lower wind speeds (<5 m s$^{-1}$) but not all.

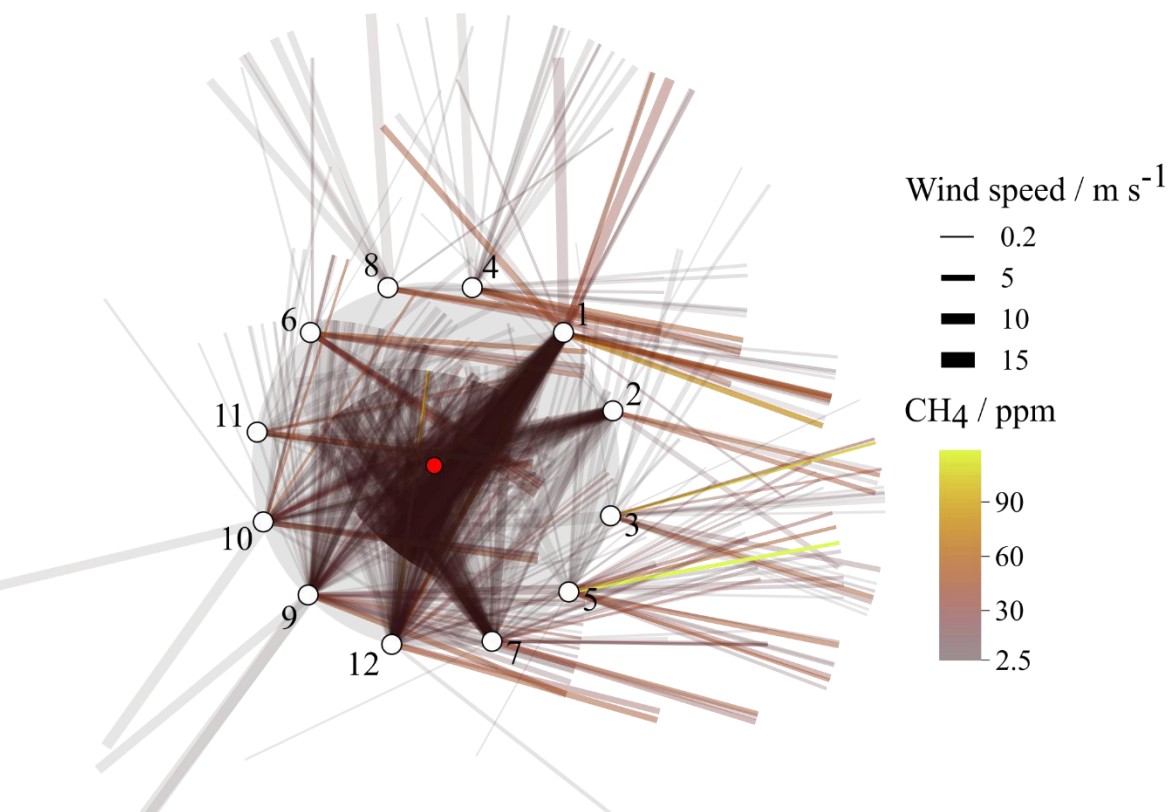

**Figure A2. Graphical representation of back trajectories (straight lines only) concurrent to detection of elevated $CH_4$ concentration**
**($CH_4$ >2.5 µmol mol$^{-1}$). The colour (and transparency) of the lines indicates the $CH_4$ concentration value (the greater the transparency, the lower the concentration). The width of the lines indicates the wind speed.**

## Appendix B: Emission quantification models

This appendix describes the three models used for emission quantification.

### Airviro

Airviro (v4.0) is an air quality management system which contains a number of dispersion modelling tools (Apertum, www.airviro.com). Airviro includes an inverse Gaussian plume dispersion model referred to as the Airviro Receptor model (Wickström et al., 2010). The Airviro Receptor model comprises two distinct modules; the source location (SL) module and the emission estimation (EE) module. These two modules can be run sequentially or separately. The Airviro Receptor model uses time series data and can be run across any time period (greater than the time resolution of the input data) for which data

is available. There must be data in the Airviro model for all receptors (i.e., sampling locations for FEDS) and meteorological stations utilised for each time step being modelled in order for the model to complete. As $CH_4$ concentration data were measured sequentially (and not continuously) at each sampling location, meteorological and $CH_4$ concentration data were both input to Airviro as hourly averages (separate data for each sampling inlet location in the case of $CH_4$). The 'timestep duration' in Airviro (e.g., hourly, 12-hourly, daily) was specified directly via a start and end time for each simulation, allowing for the

estimation of emissions over different time scales. Vertical dispersion in Airviro was modelled using input temperature data from two different heights.

Source Location (SL) module

The SL module is used to estimate the emission source location and outputs a map highlighting the regions of highest 'source probability' in which Airviro estimates the greatest likelihood of an emissions source(s). The SL module utilises a user defined

grid and cell resolution (typically 10 m × 10 m). Initially, the model places a "pseudo-source" at the centre of each grid cell and runs the Gaussian model forwards to calculate expected concentrations at various receptors (i.e., sampling locations). For each "pseudo-source", the correlation between the simulated and the measured concentration-time series is calculated. The calculations are repeated for all pseudo-sources (one for each grid cell). The horizontal distribution of the correlation coefficients is displayed as isolines multiplied by 100 to yield a 'source probability' at each location. Information from the SL

module may then be used by the user to constrain the EE module of the Airviro Receptor model.

Emission estimation module

The EE module requires the user to specify area source(s) (or emission zones) as rectangular boxes. The user also specifies an associated source height for each emission zone. The EE module assumes a constant background concentration of the target species (in this case $CH_4$). Within the EE module, Airviro is constrained to only attribute emissions to the user-specified

emission zone(s) within the model grid (unlike in the SL module). Following input of emission zones, Airviro divides each emission zone area into nine "sub-areas". Initially, the EE module assumes each sub-area has a constant, default emission and runs the dispersion model forwards to calculate the expected concentrations at the receptors (as in the SL module). The EE module performs a multiple regression analysis to simulate source contributions and optimises this with the actual measured

concentration data. Airviro outputs a best estimate of the emission rate for each emission zone, along with an associated error.

Whilst the error associated with the emission estimated by Airviro is a useful parameter, it should not be considered equivalent to a full uncertainty budget. Firstly, the model error does not account for uncertainty in the measured input parameters (either concentration or meteorology). Secondly, inverse dispersion modelling of time-series data where sources are not always well understood is a highly complex and indirect method of deriving emission rates and therefore may be expected to have a reasonably high associated uncertainty (potentially up to 50% of estimated emissions but likely highly dependent on the quality

of measurements and environmental conditions at the time of monitoring, as well as the fidelity of the model). The errors in emission rates output by Airviro are frequently less than 10% of the estimated emission rate. This may be unrealistically low. NPL is investigating the development of a robust uncertainty budget for local-scale reverse dispersion models, such as the Airviro Receptor model.

**WindTrax**

WindTrax 2.0 (ThunderBeach Scientific, www.thunderbeachscientific.com) is a Lagrangian stochastic particle model used for modelling atmospheric transport over small horizontal distances (<1 km) (Flesch et al., 1995; 2009). WindTrax is specified in the United States Environmental Protection Agency (EPA) Other Test Method 33 (OTM-33) (US EPA, 2014). OTM-33 details a standardised method for the geospatial measurement of air pollutants and remote emissions quantification, typically applied to monitoring from a mobile platform. For context, Other Test Methods are those methods which are not yet subject to the US

Federal rulemaking process but may still be used in Federally enforceable programs. WindTrax has been used to quantify $CH_4$ emissions from landfill and hydraulic fracturing (e.g., Riddick et al., 2017; Shaw et al., 2020).

Here, the $CH_4$ emission rate was estimated using the modelled advection of 50,0000 particles from the release location. A surface roughness length ($Z_0$) of 2.3 cm (equivalent to short grass) was used. Different particle numbers were examined for the impact on estimated emission rate, but different surface roughness lengths were not tested. Meteorological

and $CH_4$ concentration data were input to WindTrax as hourly averages (separately for each sampling location in the case of $CH_4$). Atmospheric turbulence was modelled using a default value for the Monin-Obukhov length. WindTrax requires the user to input a $CH_4$ background concentration (see Appendix C). The WindTrax timestep cannot be specified and is always equal to the timestep of the input data. Multiple hours of output data can be averaged to provide estimates across different time durations. WindTrax also outputs a standard deviation in emission rate for each hour of data. The model error within WindTrax

is calculated by dividing particles into 10 subgroups and assessing the standard deviation of the mean value estimated using each subgroup. As in the case of the model error output by Airviro, it is unlikely that the WindTrax model error represents a complete uncertainty budget, not least because it doesn't include uncertainty within the measured input parameters.

In general, the model used in WindTrax assumes that there is an equal number of concentration measurements (i.e., sampling locations) and unknown emission sources. If the number of concentration measurements is greater than the number

of unknown sources (as is the case in the field set up here), the simultaneous equations are solved using a least-squares fit.

**Custom Gaussian plume model**

CH$_4$ emissions were also quantified using a simple custom-built Gaussian plume analysis, similar in concept to that described in OTM-33A (US EPA, 2014). OTM-33A is applicable for ground-level emission sources located within 200 m of the receptor. OTM-33A has provision for stationary measurements made within an emission plume, and this has been utilised in several emission studies (Foster-Wittig et al., 2015; Shaw et al., 2020). Variations in wind direction move the plume around the stationary location in three dimensions over the sampling period. A point-source Gaussian analysis is then employed to derive the emission rate. For the purposes of this work, the typical approach to emission quantification prescribed in OTM-33A was adapted to the use of multiple stationary receptors. The wind direction was rotated to the frame of reference for each individual sampling inlet location, with a rotated wind direction of zero representative of a wind direction blowing over the controlled release location and directly towards the sampling inlet location. CH$_4$ concentration data were then binned into 2° bins (by rotated wind direction) and averaged. A Gaussian function was fit to the averaged concentration data.

The maximum CH$_4$ concentration output by the Gaussian function ($a1$) was used to estimate the CH$_4$ emission rate, using Eq. B1 (US EPA, 2014).

$$F = a1 \times \iint \exp -\left(\frac{y^2}{2\sigma_y^2} + \frac{z^2}{2\sigma_z^2}\right) = 2\pi \times a1 \times \sigma_y \times \sigma_z \times u \qquad \text{Eq. B1}$$

Where $F$ is the CH$_4$ emission flux, $a1$ is the maximum CH$_4$ concentration output from the Gaussian function, $u$ is the mean wind speed, and $\sigma_y$ and $\sigma_z$ are the standard deviations of the crosswind ($y$) and vertical ($z$) plume spread respectively. Included within OTM-33A is provision for determining the Pasquill atmospheric stability class during measurements using the measured standard deviation in wind direction ($\sigma_{wd}$). The stability class, along with the distance between the release point and the receptor, are in turn used to determine values for $\sigma_y$ and $\sigma_z$ from a look-up table (US EPA, 2014). Using one-minute measured values of $\sigma_{wd}$, the atmospheric stability class was calculated to be '*Extremely stable*' (Class 7) for 82%, 91%, 92%, and 94% of the duration of R1, R2, R3, and R4 respectively (see Appendix A).

Uncertainties in the emission rate were calculated by propagation of the uncertainties in each element in Eq. B1 in quadrature. When assessing the uncertainty in $\sigma_y$ and $\sigma_z$, the difference between the $\sigma_y$ and $\sigma_z$ values for Class 7 and the adjacent stability class (Class 6, '*Moderately stable*') was used as a proxy for uncertainty in $\sigma_y$ and $\sigma_z$.

A high temporal density of data is typically required to generate the required Gaussian function with this method. For this reason, minute-averaged CH$_4$ concentration data were used with the Gaussian plume model (and not hourly-averaged data as was used with both Airviro and WindTrax).

## Appendix C: Additional results

Figure C1 shows time series of emission rates estimated by Airviro and WindTrax for all tested timesteps and averaging periods, compared with the controlled releases.

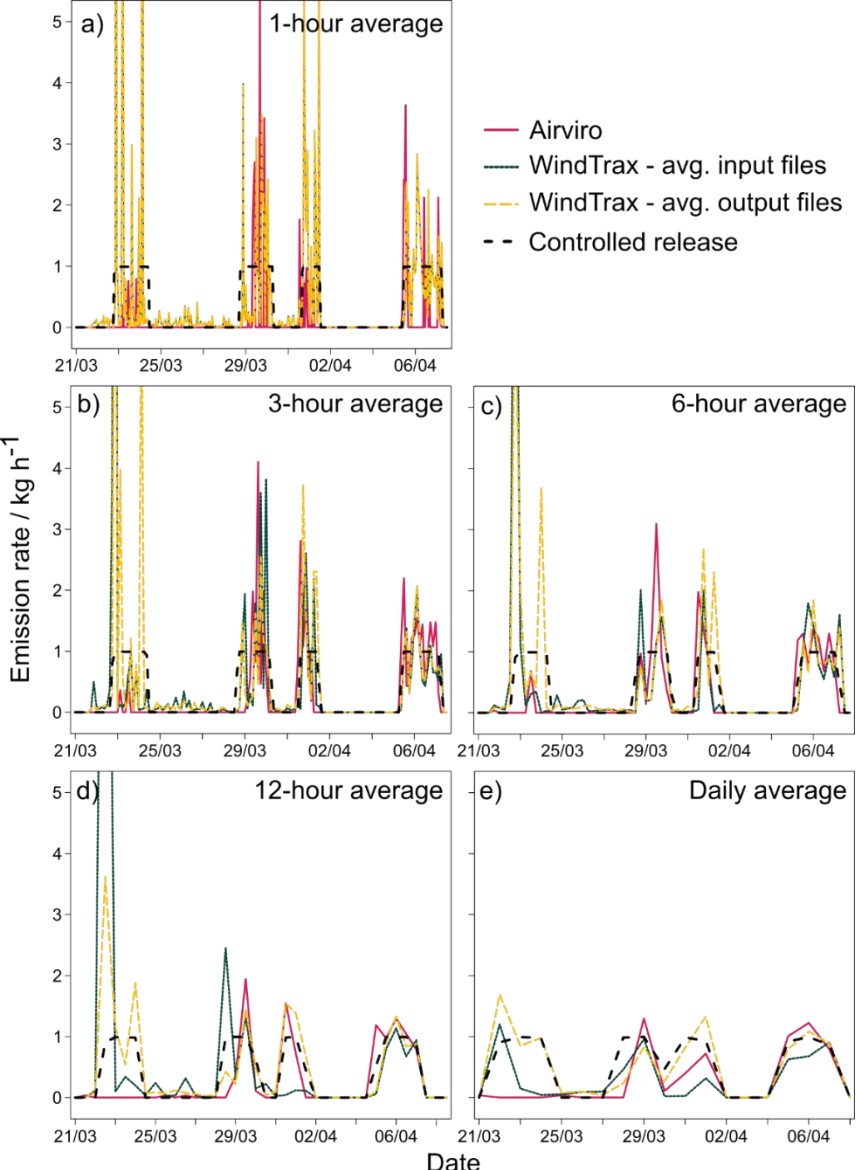

**Figure C1. Time series of time averaged emission rate results for Airviro and WindTrax (averaging either the input data or 1-hourly output results). The black dashed line shows time averaged controlled release rates. Time averaging periods used for a) hourly-average, b) 3-hour average, c) 6-hour average, d) 12-hour average, and e) daily-average. Note that the results for averaging the input and output files for hourly-averaging for WindTrax are identical.**

Figure C2 shows the impact of wind persistence on the quantified emission rates.

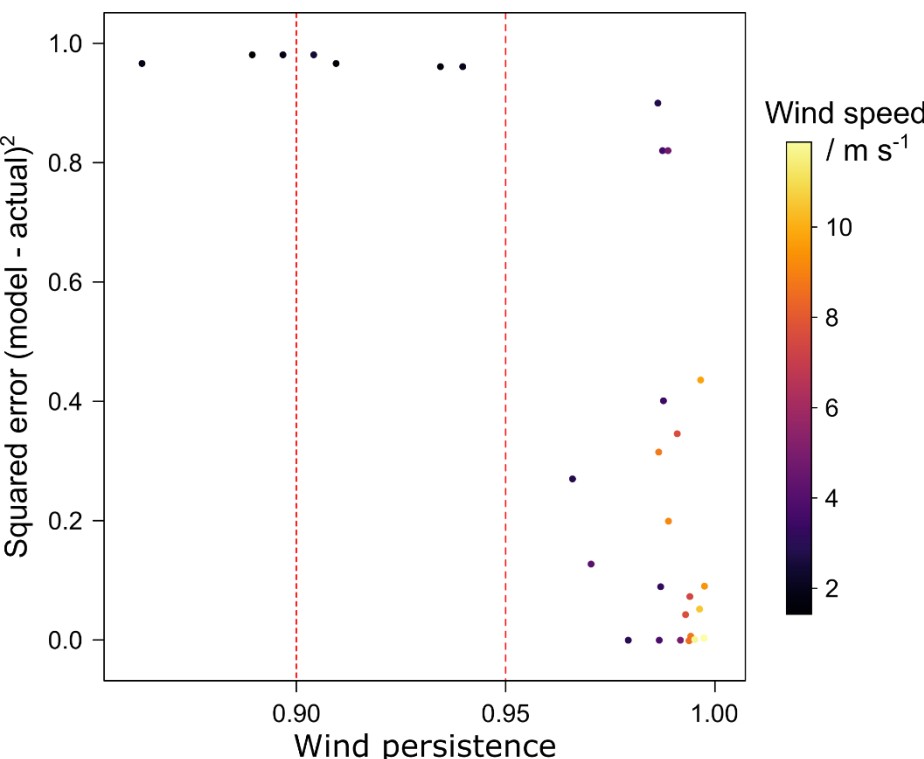

**Figure C2. The impact of wind persistence on Airviro estimated emissions, where Airviro results are represented by the square of the error between the modelled emission rate and the controlled emission rate. Data points are shown for daily and 12-hourly time averages. Data points coloured by wind speed (in m s$^{-1}$). Two wind persistence thresholds are shown by the dashed red lines, equivalent to wind persistence values of 0.95 and 0.90.**

805

Table C1 shows the same results as those in Table 4, but with periods where average hourly wind persistence was below a threshold of 0.95 (<0.95) removed. Removing periods of lower wind persistence improved the success index for Airviro, particularly for longer averaging periods, with daily and 12-hourly time steps over 90% successful at detecting the occurrence (or non-occurrence) of emissions. The Airviro bias also improved after removing periods of low wind persistence, with all averaging periods (except hourly) showing a mean bias closer to zero. The RMSE decreased somewhat for daily and 12-hourly averaging periods but increased marginally for 6-hourly and 3-hourly averaging. The only timestep which showed little-to-no improvement over including periods of low wind persistence for Airviro was the hourly averaging. This was due to the large proportion of no emissions calculated even when wind persistence was reasonably high (>0.95).

The picture for WindTrax was slightly different after removing periods of low wind persistence. The model performance actually marginally decreased in some cases. This was largely because WindTrax was able to output an emission rate even during periods of low wind persistence, where Airviro struggled.

**Table C1. Metrics comparing the Airviro and WindTrax modelled emissions rates with the controlled release emission rates whilst varying the time step (or averaging period). Periods of lower wind persistence (<0.95) were removed from the data relative to Table 4 (see Fig. C3 for contingency matrix pre- and post-application of the wind persistence threshold). Values in brackets are for a wind persistence threshold of <0.90. See Figure 8 for graphical summary.**

| Model | Time step (or averaging period) | Success index / % | Bias / kg h$^{-1}$ | RMSE / kg h$^{-1}$ |
|---|---|---|---|---|
| Airviro | Daily | 94 (85) | -0.13 (-0.26) | 0.37 (0.51) |
| | 12-hourly | 92 (86) | -0.01 (-0.09) | 0.41 (0.47) |
| | 6-hourly | 85 (83) | -0.06 (-0.08) | 0.54 (0.53) |
| | 3-hourly | 80 (79) | -0.09 (-0.10) | 0.65 (0.64) |
| | Hourly | 60 (60) | -0.31 (-0.32) | 0.77 (0.76) |
| WindTrax | Daily | 67 (67) | 0.06 (-0.01) | 0.29 (0.34) |
| | 12-hourly | 61 (64) | 0.17 (0.16) | 0.66 (0.64) |
| | 6-hourly | 57 (59) | 0.18 (0.22) | 1.05 (1.08) |
| | 3-hourly | 63 (64) | 0.17 (0.15) | 1.32 (1.27) |
| | Hourly | 68 (67) | 0.17 (0.15) | 2.19 (2.14) |

Figure C3 shows contingency matrices for both Airviro and WindTrax when using no wind persistence threshold (all data)
and when using a wind persistence threshold value of 0.95. The percentage of true positives increased for Airviro (with the
threshold applied) in all cases with different time averaging except for hourly. For Airviro, this was largely a result of the
removal of data during R1 when the wind persistence was generally below 0.95. The impact of using a wind persistence
threshold on the contingency metrics for WindTrax was much less pronounced, as WindTrax output emission rates during R1.

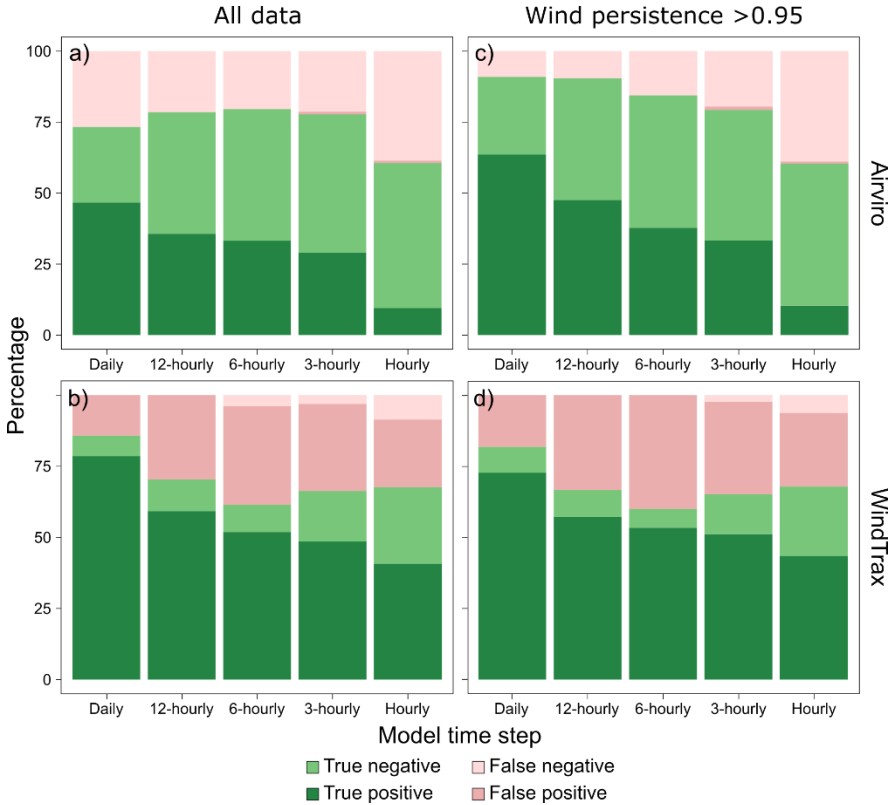

**Figure C3. Contingency matrices (percentages of true positives, true negatives, false positives, and false negatives) for Airviro (top panels) and WindTrax (bottom panels) when using all data (left panels) and when using a wind persistence threshold of 0.95 (right panels). All results shown with a lag time of 60 seconds and a plume height of 2 m.**

Table C2. Average values for wind persistence, controlled release rate (kg h$^{-1}$), and Airviro and WindTrax estimated emission rates (with and without a wind persistence threshold of 0.95 applied) over the periods covering each of the four controlled releases and under different time averaging scenarios. Note that the specific start and end dates for releases impacted the average CRF rate over different time periods; if, for a non-realistic example, the release covered the period 12:00-23:59, the average daily emission rate covering that period would be 0.5 kg h$^{-1}$ whereas the average 12-hourly rate for that period would be 1 kg h$^{-1}$.

| Time | Release number | R1 | R2 | R3 | R4 |
|---|---|---|---|---|---|
| Daily | Wind persistence | 0.94 | 0.97[a] | | 1.00 |
| | CRF emission rate (kg h$^{-1}$) | 0.96 ± 0.05 | 0.87 ± 0.23[a] | | 0.92 ± 0.07 |
| | Airviro (kg h$^{-1}$) | 0 | 0.50 ± 0.52[a] | | 1.01 ± 0.22 |
| | Airviro w/ wind pers. threshold (kg h$^{-1}$) | 0 | 0.63 ± 0.51[a] | | 1.01 ± 0.22 |
| | WindTrax (kg h$^{-1}$) | 1.17 ± 0.45 | 0.70 ± 0.46[a] | | 0.85 ± 0.21 |
| | WindTrax w/ wind pers. threshold (kg h$^{-1}$) | 1.69 | 0.81 ± 0.43[a] | | 0.85 ± 0.21 |
| 12-hourly | Wind persistence | 0.93 | 0.95 | 0.99 | 1.00 |
| | CRF rate | 0.97 ± 0.04 | 0.90 ± 0.18 | 0.81 ± 0.31 | 0.87 ± 0.20 |
| | Airviro | 0 | 0.60 ± 0.91 | 0.76 ± 0.77 | 1.06 ± 0.19 |
| | Airviro w/ wind pers. threshold | 0 | 0.80 ± 1.00 | 0.77 ± 0.76 | 1.06 ± 0.19 |
| | WindTrax | 1.80 ± 1.33 | 0.64 ± 0.54 | 1.20 ± 0.47 | 0.77 ± 0.45 |
| | WindTrax w/ wind pers. threshold | 3.62 | 0.72 ± 0.64 | 1.20 ± 0.47 | 0.77 ± 0.45 |
| 6-hourly | Wind persistence | 0.92 | 0.95 | 0.99 | 1.00 |
| | CRF rate | 0.98 ± 0.03 | 0.86 ± 0.27 | 0.88 ± 0.24 | 0.88 ± 0.22 |
| | Airviro | 0.10 ± 0.26 | 0.81 ± 1.02 | 0.84 ± 0.86 | 0.96 ± 0.44 |
| | Airviro w/ wind pers. threshold | 0 | 1.09 ± 1.05 | 0.84 ± 0.86 | 0.96 ± 0.44 |
| | WindTrax | 2.04 ± 2.59 | 0.63 ± 0.64 | 1.30 ± 1.11 | 0.84 ± 0.47 |
| | WindTrax w/ wind pers. threshold | 2.41 ± 3.26 | 0.84 ± 0.60 | 1.30 ± 1.11 | 0.84 ± 0.47 |
| 3-hourly | Wind persistence | 0.92 | 0.96 | 0.99 | 1.00 |
| | CRF rate | 0.97 ± 0.05 | 0.94 ± 0.15 | 0.92 ± 0.19 | 0.93 ± 0.17 |
| | Airviro | 0.07 ± 0.18 | 0.77 ± 1.18 | 0.90 ± 1.00 | 1.03 ± 0.58 |
| | Airviro w/ wind pers. threshold | 0.04 ± 0.12 | 0.83 ± 1.21 | 0.90 ± 1.00 | 1.03 ± 0.58 |
| | WindTrax | 1.88 ± 3.50 | 0.72 ± 0.76 | 1.52 ± 1.21 | 0.90 ± 0.49 |
| | WindTrax w/ wind pers. threshold | 2.00 ± 3.85 | 0.78 ± 0.76 | 1.52 ± 1.21 | 0.90 ± 0.49 |
| Hourly | Wind persistence | 0.92 | 0.98 | 0.99 | 1.00 |
| | CRF rate | 0.97 ± 0.14 | 0.97 ± 0.09 | 0.97 ± 0.07 | 0.95 ± 0.17 |
| | Airviro | 0.05 ± 0.18 | 0.47 ± 1.25 | 0.17 ± 0.33 | 0.40 ± 0.83 |
| | Airviro w/ wind pers. threshold | 0.03 ± 0.16 | 0.50 ± 1.28 | 0.18 ± 0.34 | 0.40 ± 0.83 |

| | | | | |
|---|---|---|---|---|
| WindTrax | $1.96 \pm 6.20$ | $0.78 \pm 1.03$ | $1.67 \pm 1.86$ | $0.94 \pm 0.70$ |
| WindTrax w/ wind pers. treshold | $1.96 \pm 6.65$ | $0.82 \pm 1.05$ | $1.78 \pm 1.92$ | $0.94 \pm 0.70$ |

[a] Release 2 (R2) ended on the same day that Release 3 (R3) began. Thus, if using daily averaging (between 00:00 and 23:59), the two releases could not be separated in time. This was not a problem for shorter averaging periods (e.g., 12-hourly) as R2 finished in the morning (before 12:00) and R3 began in the afternoon (after 12:00).

845

The lag time refers to the period of time over which data was excluded following switching between sampling inlet locations. There was unlikely to be instantaneous measurement of air sampled from the new location due to a lag in the flushing time of the optical cell. However, the longer the chosen lag time, the more data invalidated and the smaller proportion of quality data available for analysis and model interpretation.

Airviro and WindTrax were tested using different lag periods to understand if this had any impact on quantified emission rates. Varying the lag time seemed to have only a minor effect on the success index for both models (see Table C3). Figure C4 shows the bias and RMSE scores achieved whilst varying the sampling lag time for both Airviro and WindTrax. Shorter lag times yielded smaller biases for Airviro but larger values for RMSE, indicating that an intermediate value for lag time may be optimal for Airviro. The picture for WindTrax was more mixed with no clear trend. The lowest bias and RMSE values for WindTrax were observed when using a lag time of 120 s, with higher bias and RMSE scores observed when using a lag time of 75 s. Overall, varying the length of the lag period between sampling inlets had only a minor impact on the magnitude of emission rates estimated by both models.

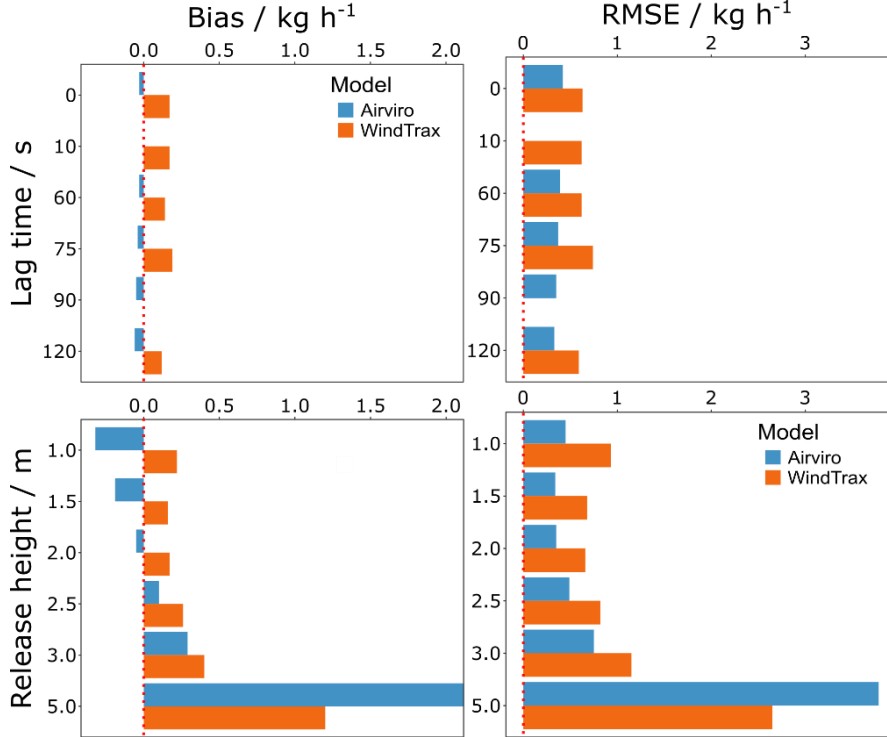

**Figure C4. Graphical summary of bias (left column) and RMSE (right column) for Airviro (blue) and WindTrax (orange) under model simulations with different lag times (top row) and release heights (bottom row). The lag time refers to the length of time that CH$_4$ concentration data was invalidated following switching of sampling inlets (to preclude the possibility of measuring air sampled from the previous sampling inlet). The release height (or plume height) refers to the height of the simulated release point. Ideal values are indicated by red dashed lines (success index = 100%; bias = 0 kg h$^{-1}$; RMSE = 0 kg h$^{-1}$). See Table C3 for values.**

Both Airviro and WindTrax require the user to specify the height at which gas was emitted. The CRF released $CH_4$ from a height of 2 m above ground level. However, knowledge of the release height may not be known in a real-world scenario, particularly if the emission source is unknown or if there are multiple emission sources in proximity. Figure C4 shows the sensitivity of both models to the release height (when using 12-hourly time periods and a wind persistence threshold of 0.95; all other variables constant). For context, the true release height of the emission plume was 2.0 m. See Table C3 for success index results.

Both models performed better overall when the release height was set to the actual value of 2.0 m. Airviro and WindTrax both yielded lower bias and RMSE values for simulations at 2.0 m, although some bias and RMSE scores were marginally lower for simulations with plume heights within 0.5 m of the true release height. The bias and RMSE values for both models increased as the simulated plume height was set at greater distances from the true release height, with the worst performances when the simulated plume was set at 5.0 m. The results therefore imply that small discrepancies in emission plume height (±0.5 m in this case) may not substantially alter the final results, but that discrepancies greater than this value (e.g., ≥±1.0 m) can result in substantial model errors. More controlled releases tests at different heights would need to be undertaken to evaluate the influence of plume height in greater detail.

Table C3 shows the success index, bias, and RMSE results for Airviro and WindTrax whilst varying the lag time (in seconds) and the release height (in m). See Fig. C4 for a graphical representation of these results. Varying the lag time (the length of time over which $CH_4$ concentration data was invalidated following switching of sampling inlets) had only a marginal impact on the success index, bias, and RMSE results for both models. A lag time of between 60 and 90 seconds appeared to be optimal. The model setting for release height had a much more substantial impact on the quantified emissions. The results for Airviro showed the model switched bias, with Airviro underestimating emissions for release heights at or below the true release height, but overestimating emissions for release heights greater than the true release height.

**Table C3. Metrics comparing the Airviro and WindTrax modelled emissions rates with the controlled release emission rates whilst varying different parameters. The lag time refers to the length of time that $CH_4$ concentration data was invalidated following switching of sampling inlets (to preclude the possibility of measuring air sampled from the previous sampling inlet). The release height (or plume height) refers to the height of the release point.**


| Model | Lag time / s | Success index / % | Bias / kg h$^{-1}$ | RMSE / kg h$^{-1}$ |
|---|---|---|---|---|
| Airviro | 0 | 88 | -0.03 | 0.42 |
| | 60 | 92 | -0.03 | 0.39 |
| | 75 | 92 | -0.04 | 0.37 |
| | 90 | 92 | -0.05 | 0.35 |
| | 120 | 92 | -0.06 | 0.33 |
| WindTrax | 0 | 64 | 0.17 | 0.63 |
| | 10 | 64 | 0.17 | 0.62 |
| | 60 | 61 | 0.14 | 0.62 |
| | 75 | 64 | 0.19 | 0.74 |
| | 120 | 64 | 0.12 | 0.59 |
| **Model** | **Release height / m** | **Success index / %** | **Bias / kg h$^{-1}$** | **RMSE / kg h$^{-1}$** |
| Airviro | 1.0 | 92 | -0.32 | 0.45 |
| | 1.5 | 92 | -0.19 | 0.34 |
| | 2.0 [a] | 92 | -0.05 | 0.35 |
| | 2.5 | 88 | 0.10 | 0.49 |
| | 3.0 | 88 | 0.29 | 0.75 |
| | 5.0 | 71 | 2.21 | 3.78 |
| WindTrax | 1.0 | 61 | 0.22 | 0.93 |
| | 1.5 | 61 | 0.16 | 0.68 |
| | 2.0* | 61 | 0.17 | 0.66 |
| | 2.5 | 61 | 0.26 | 0.82 |
| | 3.0 | 56 | 0.40 | 1.15 |
| | 5.0 | 50 | 1.20 | 2.65 |

[a] True release height.

Table C4 shows the results when varying the number of active sampling inlets used for input into Airviro and WindTrax simulations. Figure C5 also shows schematic configurations of the chosen sampling inlet locations around the CRF release point location for each model scenario.

**Table C4. Metrics comparing the Airviro and WindTrax modelled emissions rates with the controlled release emission rates whilst varying the number of active inlets used for input of CH₄ concentration data. See Fig. C5 for schematic configurations of inlet locations.**

| Scenario | Active sampling inlets | Success index / % | | RMSE / kg h⁻¹ | |
|---|---|---|---|---|---|
| | | Airviro | WindTrax | Airviro | WindTrax |
| 12a | All 12 | 92 | 61 | 0.35 | 0.64 |
| 11a | Excluding no. 1 | 83 | | 0.50 | |
| 11b | Excluding no. 12 | 88 | | 0.40 | |
| 11c | Excluding no. 6 | 92 | | 0.36 | |
| 11d | Excluding no. 2 | 92 | | 0.38 | |
| 11e | Excluding no. 9 | 92 | | 0.36 | |
| 11f | Excluding no. 8 | 92 | | 0.37 | |
| 11g | Excluding no. 5 | 92 | | 0.39 | |
| 9a | Excluding nos. 1, 7, 11 | 79 | 61 | 0.56 | 1.37 |
| 9b | Excluding nos. 2, 6, 12 | 88 | 67 | 0.41 | 0.58 |
| 9c | Excluding nos. 3, 8, 9 | 50 | | 0.65 | |
| 9d | Excluding nos. 4, 5, 10 | 50 | | 0.67 | |
| 8a | Excluding nos. 3, 4, 11, 12 | 88 | 61 | 0.56 | 1.51 |
| 8b | Excluding nos. 1, 5, 6, 9 | 83 | 56 | 0.53 | 1.59 |
| 8c | Excluding nos. 2, 7, 8, 10 | 88 | 56 | 0.51 | 0.85 |
| 6a | Nos. 1, 3, 7, 8, 9, 11 | 88 | 67 | 0.68 | 0.82 |
| 6b | Nos. 2, 4, 5, 6, 10, 12 | 79 | 61 | 0.61 | 1.73 |
| 3a | Nos. 1, 7, 11 | 75 | | 0.92 | |
| 3b | Nos. 4, 5, 10 | 58 | 43 | 0.65 | 0.46 |
| 3c | Nos. 3, 8, 9 | 58 | 64 | 0.75 | 0.43 |
| 3d | Nos. 2, 6, 12 | 71 | | 0.56 | |
| 1a | No. 1 | 57 | | 2.37 | |
| 1b | No. 12 | 44 | | 0.67 | |
| 1c | No. 6 | 51 | | 0.59 | |
| 1d | No. 2 | 43 | | 0.67 | |

| | | | |
|---|---|---|---|
| 1e | No. 9 | 53 | 0.64 |
| 1f | No. 8 | 62 | 0.63 |
| 1g | No. 5 | 49 | 0.65 |


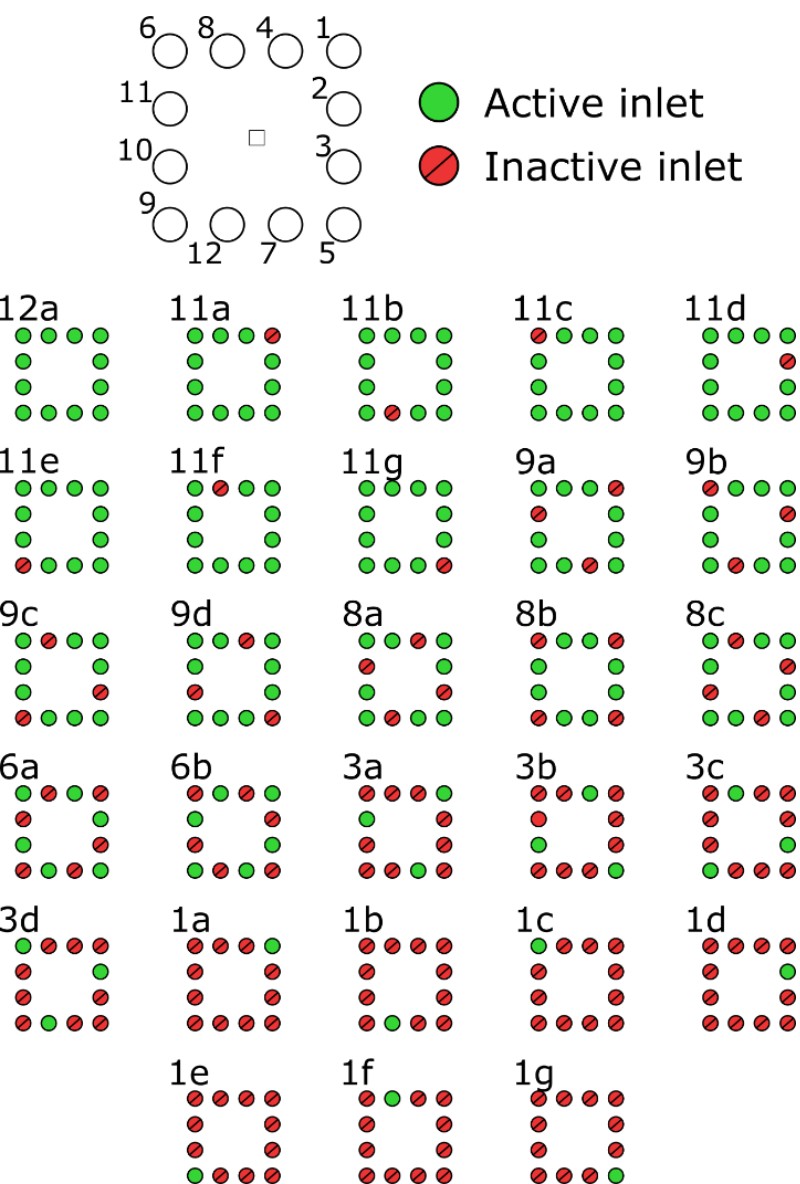

**Figure C5. Schematics of scenarios in which differing configurations of sampling inlet locations were input for modelling scenarios (see Table C4). See also Fig. 2.**

WindTrax requires the user to specify a value for the local $CH_4$ background concentration. The $CH_4$ background concentration can be a difficult value to prescribe accurately, especially in the presence of extraneous emission sources and also in the context of long-term monitoring over which the background might be expected to vary with season (e.g., Shaw et al., 2019). Airviro has no user-specified setting for the $CH_4$ background and instead calculates its own background value internally.

Table C5 shows the results of varying the $CH_4$ background setting for WindTrax between 0 and 2 µmol mol$^{-1}$, with 2

µmol mol$^{-1}$ representative of a typical UK background at the time of publication. A lower $CH_4$ background value resulted in WindTrax overestimating emissions, as might be expected. WindTrax performed best when the $CH_4$ background was set to a value which could be considered reasonably representative of the true background (the 5[th] percentile of $CH_4$ mole fractions measured during the campaign was 2.002 µmol mol$^{-1}$). More simulations varying the background value across a range of actually feasible background methane concentrations (e.g., 1.90 – 2.10 µmol mol$^{-1}$) would be instructive for determining the

sensitivity of WindTrax to the choice of background value.

**Table C5. Metrics comparing WindTrax modelled emissions rates with the controlled release emission rates whilst varying the $CH_4$ background concentration.**

| Model | $CH_4$ background concentration / µmol mol$^{-1}$ | Success index / % | Bias / kg h$^{-1}$ | RMSE / kg h$^{-1}$ |
|---|---|---|---|---|
| WindTrax | 0 | 50 | 3.85 | 4.88 |
| | 1 | 50 | 2.00 | 2.51 |
| | 2 | 57 | 0.05 | 0.55 |


Airviro does not have capability to simulate single point emission sources and instead requires the user to designate an emission source zone from which emission rates are then calculated. WindTrax, on the other hand, allows for the user to specify the location of known point sources. Figure C6 shows a number of different emission zones which were tested with Airviro to assess the sensitivity of the model to the size of emission zone. It should be noted that the default emission zone used for all other scenarios was Emission Zone 3.

The success index decreased with increasing size of the emission zone, especially beyond the circle of sampling locations. When the emission zone was kept to mostly within the circle of sampling locations, the bias was negative. However, the bias was positive if the emission zone was extrapolated outside of the circle of sampling inlets. RMSE was lower for emission zones within the circle, and greater for emission zones that extended beyond the circle. This implies that Airviro was best able to capture emissions when multiple sampling locations surrounded the emission source.

**Table C6. Metrics comparing Airviro modelled emission rates with the controlled release emission rates whilst varying the emission zone (see Fig. C6).**

| Model | Emission zone | Success index / % | Bias / kg h$^{-1}$ | RMSE / kg h$^{-1}$ |
|---|---|---|---|---|
| Airviro | 1 | 92 | -0.20 | 0.39 |
| | 2 | 92 | -0.15 | 0.33 |
| | 3 | 92 | -0.05 | 0.35 |
| | 4 | 65 | 0.58 | 1.17 |
| | 5 | 50 | 1.05 | 2.07 |

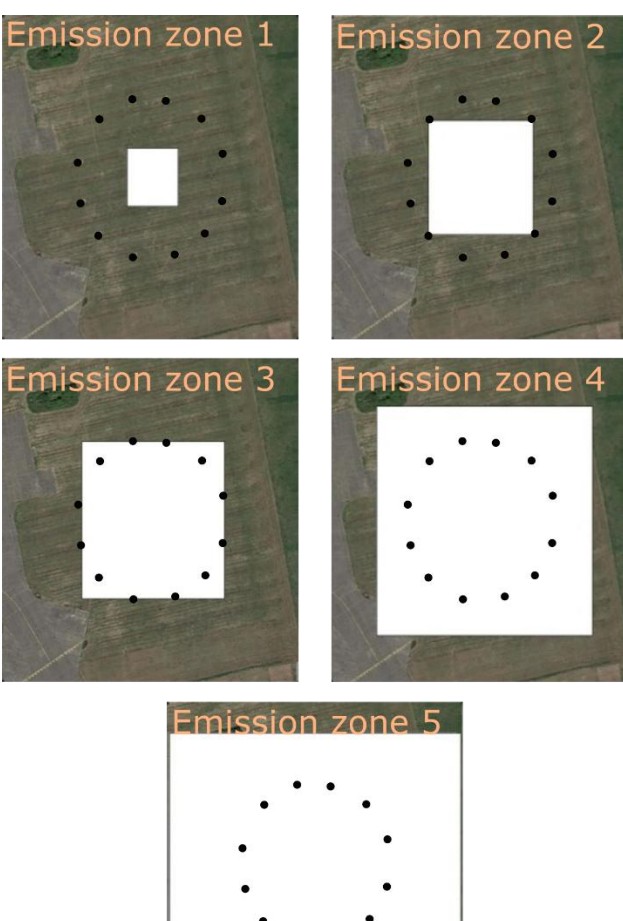


Figure C6. Configuration of five different emission zones (white areas) tested with Airviro. The black dots indicate the locations of the sampling inlets (see Fig. 2). Emission zone 3 was used for all other Airviro simulations in this work. Satellite imagery: Google, ©2020 Maxar Technologies.

WindTrax allows the user to specify the number of particles modelled for dispersion for each time step. Increasing the number of particles theoretically increases the accuracy of dispersion modelling but this comes at the expense of increased computational power and calculation time.

Table C7 shows the results of varying the particle number for WindTrax between 5,000 particles and 100,000 particles. The simulation with the fewest particles (5,000) resulted in a negative bias whereas all other simulations yielded

positive biases. The simulation with 5,000 particles also resulted in the lowest value for RMSE. The simulation with 10,000 particles gave the greatest success index and lowest bias but one of the higher values for RMSE. Increasing the particle number beyond 30,000 particles led to no improvement in success index and only marginal improvements in bias. Increasing the particle number beyond 20,000 led to a general decrease in RMSE up to 50,000 particles. 50,000 particles were used in most other WindTrax simulations in this report as it was seen as a compromise between reasonable metric scores and short

calculation time.

**Table C7. Metrics comparing WindTrax modelled emissions rates with the controlled release emission rates whilst varying the particle number.**

| Model | Particle number | Success index / % | Bias / kg h$^{-1}$ | RMSE / kg h$^{-1}$ |
|---|---|---|---|---|
| WindTrax | 5,000 | 75 | -0.34 | 0.52 |
| | 10,000 | 79 | 0.07 | 0.89 |
| | 20,000 | 57 | 0.22 | 0.97 |
| | 30,000 | 61 | 0.21 | 0.81 |
| | 40,000 | 61 | 0.19 | 0.71 |
| | 50,000 | 61 | 0.17 | 0.66 |
| | 100,000 | 61 | 0.19 | 0.70 |


**Data availability**

Data may be made available upon request.

**Author contributions**

Author contributions using CRediT (Contributor Roles Taxonomy).

JTS: Formal analysis, Methodology, Validation, Visualization, Writing – original draft preparation.

NH: Conceptualization, Data curation, Formal analysis, Investigation, Methodology, Validation, Writing – original draft preparation.

JC: Conceptualization, Data curation, Formal analysis, Investigation, Methodology, Validation.

DEB: Formal analysis, Visualization, Writing – original draft preparation.

JR: Data curation, Investigation, Software.

JH: Investigation, Validation, Writing – original draft preparation.

NY: Investigation, Validation, Formal analysis.

DB: Conceptualization, Methodology, Supervision.

FI: Conceptualization, Methodology, Supervision.

RR: Conceptualization, Methodology, Project administration, Supervision, Writing – original draft preparation.

**Competing interests**

The FEDS system is a commercial property owned and operated by NPL.

**Acknowledgements**

We would like to thank all other staff at NPL who have contributed to the development, deployment, and operation of FEDS since its conception. This includes David Butterfield, who helped with the original build including inlet connections and networking infrastructure to allow for remote operation, and Tom Gardiner and Fabrizio Innocenti, who assisted with the conceptualisation of FEDS. Other NPL staff who have helped to develop, deploy, and operate FEDS include Chris Bradshaw, Andy Connor, Andrew Finlayson, Alex Hazzard, Alice Hirons, James Frost, Alex McColl, Linh Nguyen, Adam Howes, and Jim Keene. NPL would also like to acknowledge the initial support for the development of FEDS from the European Institute of Innovation and Technology (EIT) and the National Grid (now National Gas Transmission) as part of the Fugitive Methane Emissions (FuME) and Monitoring of Real-time Fugitive Emissions (MoRFE) projects.

**Financial support**

Funding for this work was supported as part of the National Measurement System Programme delivered by the Department for Science, Innovation and Technology (DSIT).

985

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
