# Peer review of "Assessment of the Fugitive Emission Distributed Sampling (FEDS) system: A mobile, multi-inlet system for continuous emissions monitoring"

_EGUsphere, 2025_

## Author Comment (AC2)

**Authors' response to reviewer comments on egusphere-2025-1451**

The authors would like to thank the reviewers for their comments in support of this publication. All reviewer comments are collated below in *italics* with corresponding author responses in blue text. Additions or amendments to the original manuscript are further highlighted by underlining.

Reviewer 1

*General: The work presented here feels a little "work in progress" rather than a true validation of the system. Whilst it is a useful step in the validation process of FEDS, I am a little concerned that it doesn't demonstrate the capabilities in a truly rigorous way in comparison to the intended use scenarios and that a follow-up piece of work is needed. I would request that the term "validation" is removed from the title as I think more is needed – maybe "first assessment" is more appropriate?*

*I believe this work is worth publishing after revisions, as it lays the groundwork for the FEDS system. A part II controlled release study paper would be welcomed – possibly alongside the first site level measurements?*

*There are a number of significant improvements and increase in scope that I would like the authors to consider. Some should be feasible within the boundaries of the current dataset, but others I would like them to be mindful of as they design their next controlled release experiments to assess and improve the system.*

The authors would like to thank the reviewer for their time spent reviewing this publication and their constructive comments. We agree that more work is needed to comprehensively validate such a monitoring system across a range of emission scenarios but recognise that this requires time, effort, and funding. We have amended the title of the of the paper to reflect the reviewers concerns on validation. The paper title has been amended to:

"Assessment of the Fugitive Emission Distributed Sampling (FEDS) system…"

Further references to "validation" throughout the manuscript have been removed (where necessary) and reworded (often by switching with the word "assessment").

We are certainly happy to take on board the reviewers' recommendations for future controlled release experiments that we undertake to further validate the FEDS system.

1. *The work should be as rigorously compared to other systems attempting to achieve the same / similar goals. The work of the Stanford and Colorado State Universities that have done significant work assessing continuous monitoring system performance should be heavily cited and compared as directly as possible through the creation of comparable performance metrics. As it currently stands, I have no direct reference in the paper as to whether this system performs well, averagely or poorly compared to other systems.*

We have added a new section 3.5 to the manuscript covering "Comparison with other continuous monitoring solutions". The section is too long to copy out here so should be referred to in the revised manuscript.

It should be noted that some of the metrics reported in work from Stanford or Colorado State Universities differ from those in this work. This was primarily due to the way in which

emissions were reported. FEDS emissions were looked at over different time periods for the same releases (i.e., hourly, 3-hourly, 6-hourly etc.) whereas other continuous monitoring solutions were evaluated on a release-by-release basis.

2. *The use of the models chosen (and why not others) is not terribly well justified. I would have liked to have seen a comprehensive look at the possible choices and then reasoning for using these and not others. There are solution possibilities using tools such as GRAL (https://gral.tugraz.at/) or even an algorithmic approach to quantification such as that being proposed by LBL for abandoned oil and gas infrastructure assessment (https://egusphere.copernicus.org/preprints/2025/egusphere-2025-344/egusphere-2025-344.pdf)*

The reviewer correctly points out that the models used in this work are not the only models available for reverse dispersion modelling at local scales. There are a limited number of other models for local scale dispersion available, such as GRAL. There are two popular model types for simulating atmospheric transport at the local scale. Models typically use either Gaussian assumptions of plume dispersion or use particles to simulate atmospheric flow and turbulence (Lagrangian stochastic models). For this work, the authors wanted to compare the results of one of each type of model. Airviro (a Gaussian model) was selected following NPL company policy and procurement procedures, matching the needs of the FEDS system to the capabilities of the model. Limited funding for both model purchasing and resource training were considered. WindTrax was selected over GRAL (both are Lagrangian models with potentially similar approaches) due to user familiarity and because of a number of other publications in the literature which used WindTrax for modelling methane dispersion.

Incorporating results from additional models into this submission is, unfortunately, beyond the funding scope of this work. Whilst models like GRAL are free to access, they still require substantial training to use and significant resourcing to develop data input protocols and quality assurance. We do have plans to carry out a more rigorous assessment of multiple site-scale (<1 km) dispersion models using FEDS data (from this and other fieldwork). We hope to use models such as GRAL alongside Airviro and WindTrax, and explore the possibility of other models (that typically operate at larger scales) such as FlexPART etc.

To this end, we have added the following sentences to Section 2.4:

"These models were chosen following an internal commercial review of available options (of both software and freeware) assessing model usability, availability, support, price, operating principle (Gaussian, Lagrangian etc.). At least one of each type of common approach to modelling atmospheric transport (Gaussian, Langrangian) was chosen so as to compare the two methods.  It should be noted that other modelling options are available but testing all available models was beyond the scope of this work."

3. *There is no "non-model" baseline to compare to. Given the simplicity of the set up I would have liked to have seen a simple analysis of the results identifying when a concentration is elevated above background by a certain threshold and then plotting a back trajectory line from the node using the measured wind fields, and using that to determine source location and provide a comparative success index without the need for a model.*

The below plot achieves what we understand the reviewer to be asking for. Here, instances where $CH_4$ concentration were greater than 2.5 ppm (indicating enhancement above

the background of ~2 ppm) were plotted with a straight line showing the back trajectory from the sampling location (using the concurrent wind direction). The majority of trajectories did pass close by to the CRF emission source although there were many instances where trajectories were away from the central emission source. Some of these instances where wind direction was away from the CRF were associated with lower wind speed (<5 m s$^{-1}$). This may indicate more complex atmospheric mixing conditions.

This figure has been added to Appendix A along with a short explanation.

[Figure]

4. *I wonder if there is room for improvement on the inlet design? There a plenty of design set ups that could have borrowed heavily from the eddy covariance community that have utilized fast 10 Hz instruments with rapid throughput of gas which would massively reduce the dead analysis period that needs to be removed and allow much faster switching between nodes. I would encourage this to be investigated and some spreadsheet calculations to work out if this is worth redesigning for in the future.*

The authors agree that there is definitely room for improvement. The FEDS system is intended to be a continuously improved upon resource and as such will undergo further refinement as and when funding is available. The use of a 10 Hz analyser may improve the throughput of gas through the optical cell reducing the sampling dead time when sampling locations are switched. Such an improvement would need to be balanced against other potential improvements, at least in the near future, but we thank (and welcome) the reviewer for their suggestion.

We have added a sentence to the conclusion mentioning this as a possible future improvement:

"Further improvements to the FEDS system may be realised via the use of gas analysers with increased cell flushing times (such as the >10 Hz analysers often used for eddy covariance measurements) to reduce the sampling lag time when switching between sampling locations."

5. *Somewhat adding to the previous point, the low time resolution of the system (one measurement period per node per hour) is very limiting and will cause severe problems when trying to scale up to sites with intermittent sources or variable emission rates. It may be okay for purely providing an overarching emission estimate from long duration deployment, but without the ability to also act as a mitigation tool the role of it as a method may be very limited in the context of continuous monitoring options.*

We agree with the reviewer that the low time resolution of the FEDS system (when viewed on a per sampling location basis) is a limitation. The possibility of missing intermittent sources and the challenge of measuring sources with variable emissions is a limitation of many monitoring options (Connor et al., 2024). So-called "snapshot" systems such as tracer gas dispersion or using unmanned aerial vehicles suffer from similar limitations in that their methodologies are only applicable to emissions at the time of monitoring (and may therefore miss an intermittent source) and they are labour-intensive to deploy long-term to monitor variable sources. The FEDS system offers a continuous monitoring option that is generally labour-light.

We have added additional discussion to the introduction to capture some of this:

"The labour-intensity of performing mobile measurements means options such as the tracer gas dispersion method (e.g., Mönster et al., 2019; Shah et al., 2025) or those utilising unmanned aerial vehicles (e.g., Shaw et al., 2021) are typically used for so-called "snapshot" monitoring and may therefore miss intermittent emission sources (if not measuring at the time of emission) or have limited capacity for determining variability in emissions over time (Connor et al., 2024)."

6. *As you have identified, understanding the wind is key. For future work, I would recommend siting a small 2D sonic at each node so that exact wind information can be obtained for each location. For any site with topography or structures, quite dramatic changes in wind can be seen across a site due to shielding and slope effects which would be important information for the modelling.*

Again, we thank the reviewer for their suggestion and agree that increased resolution of wind data could improve the accuracy of dispersion modelling, particularly as the reviewer says, in those scenarios where the wind field might be complex due to terrain or obstructions. Trialling advanced wind options would be of interest, especially if combined with computational flow models.

We have added a sentence to the conclusions to capture this:

"Additionally, collecting more wind data (e.g., by including a wind anemometer at each sampling location) may substantially improve the modelling of the wind field, which is key for accurate emission estimation and source localisation (particularly in complex environments)."

7. *The potential improvement of fully utilizing the atmospheric stability is mentioned in the paper, but not followed up. It seems that adding this analysis in is within the remit of the work as it should demonstrate whether this is something that should be included in future analysis?*

This is something that we have trialled using some other data which will be reported in the future. Our findings have shown that a neutral stability class (Paquill class D) performs well (best agreement with other independent monitoring methods deployed simultaneously) when

averaged over periods of time greater than 24 hours (thereby averaging out different atmospheric stability between day and night). Results from this will be published in future work.

8. *Why does the performance of the two models vary so significantly in terms of false positive detections? From an operator perspective this is a key metric as no one wants to send staff out to fix things that don't exist. This should have more detail around it and understand why this is occurring. This should be considered a significant problem for the windtrax solution.*

The reviewer makes a very important point and this really exemplifies why quality assurance (or greater scrutiny) of data is needed before analysis. Identifying periods of poor wind conditions (which correlates with a greater chance of false positives) may reduce the number of false positive detections. This, of course, leads to further questions about the frequency of poor wind conditions and the lack of ability to reliably detect emissions during these times.

It should be noted that the FEDS system was primarily designed as a long-term solution to quantify emissions from facility-scale sources and is not necessarily the best solution to detect and locate small leaks at the component level (thereby replacing LDAR). Whilst we are sure that false positive detection rates are a key metric to facility operators in order to efficiently and cost-effectively fix leaks, this is not the primary function of the FEDS and any source localisation provided by FEDS would be approximate (at a unit-level scale not component).

9. *There is a hint in the conclusions that low cost sensors were also trialed – I would love to see the data on this and some more details as to how it was set up. A null result is still very useful and may help others to not continue down a similar path of method.*

Considerable effort was made to deconvolute and calibrate the low-cost sensor data due to potential external interferences (from non-methane concentration) and inter-sensor variability. However, we do not have the confidence to incorporate this data into the FEDS analysis at this time. We have removed the sentence referring to the low-cost sensors from the conclusions to preclude the possibility of other readers searching for these results in the main text.

*Detailed line by line:*

*Abstract:*

*L15: Add some details of the CR - number of sources, release rate ranges, what is being simulated.*

The abstract has been amended to include the following information:

"The CRF was used to simulate a simple single-point methane emission source with constant release rate of 1 kg h$^{-1}$ over four separate experiments."

*L19: Without knowledge of the true emission rate these values are a little meaningless. If the emission rate is 100kg hr, then these are wonderful. If it is 2kg hr, less so… Maybe replace throughout with % of true emission.*

We hope that by including the true emission rate in the abstract (see above) that this issue is resolved. We have also clarified the RMSE errors equivalent relative error in the text.

*L21:23: Feels more like discussion that abstract.*

This sentence has been removed from the abstract.

*Main Text:*

*L31: efficient, accurate and transparent monitoring*

Amended.

*L41: Discuss snapshot vs continuous*

The following text has been added to the introduction:

"The labour-intensity of performing mobile measurements means options such as the tracer gas dispersion method (e.g., Mönster et al., 2019; Shah et al., 2025) or those utilising unmanned aerial vehicles (e.g., Shaw et al., 2021) are typically used for so-called "snapshot" monitoring and may therefore miss intermittent emission sources (if not measuring at the time of emission) or have limited capacity for determining variability in emissions over time (Connor et al., 2024). On the other hand, stationary sampling can provide capacity for reduced labour efforts (as sensors can be left in place) with the benefit of potential for continuous monitoring."

*L46: Put in context of reporting requirements such as the new EU regulations or voluntary programmes such as OGMP2.0? These are already in place and have specific requirements around needs.*

This is a good suggestion. We have added the following sentences to the Introduction:

"Measurement-based approaches to $CH_4$ emission reporting are now being prioritised over (or alongside) site-level estimation via activity data through programmes such as the Oil and Gas Methane Partnership 2.0 (OGMP2.0) and within recent regulatory requirements in the European Union (i.e., EU 2024/1787, 2024). OGMP2.0 requires transparent and verified site-level measurements to achieve its Gold Standard whilst EU 2024/1787 requires mandatory measurement, reporting, and third-party verification of site-level emissions."

*L74: The CR needs to be better described here. Is it fully blind, single point or multiple point etc…?*

The controlled release is described in detail in the Methodology section. We have added the following text to the Introduction:

"Multiple known and controlled releases of $CH_4$ (non-blind, single point emission source) were used to evaluate and assess…"

The following text was added to the Methodology section:

"Whilst the release rate was known prior to analysis, all data analysis was performed independent of the knowledge of the release rate (i.e., knowledge of the release was not used to influence analysis)."

*L79: I wouldn't consider custom Gaussian plume out of reach of commercial teams.*

This was intended to reflect that commercial teams may be more likely to use commercially-available software solutions rather than pursue building their own custom

software. However, the reviewer is correct that this may not always be the case, and many commercial teams do build their own analysis software. We have removed the text at the end of the Introduction pertaining to this.

*L83: Specify what is considered high-performance*

The manufacturer-reported specifications for the gas analyser are now reported in Appendix A (Table A1). High performance here refers generally to the precision, accuracy, range, and resolution of the analyser when compared with lower cost sensors. By way of a comparison against a generic low-cost sensor, we have added the following text to the Appendices alongside Table A1:

"The specifications of the LGR FMEA may be compared with lower cost sensors such as the Cubic SJH-5A Methane Sensor (which uses non-dispersive infrared (NDIR) for $CH_4$ detection), or the Gazomat CATEX™ 3 (a catalytic combustion sensor). Such low-cost sensors may be preferred for some applications but at the expense of performance and quality of data. The Cubic SJH-5A has a (manufacturer-reported) $CH_4$ measurement range of 0%-5% (by volume), accuracy of ±0.06% in the range 0%-1% (by volume), measurement resolution of 0.1%, and response time of <25 seconds. The measurement resolution is particularly poor when compared with the LGR FMEA. The specifications of the Cubic SJH-5A sensor mean that detection and identification of all but the largest emission events might be challenging. The CATEX™ 3 has a reported lower detection limit for $CH_4$ of 100 $\mu$mol mol$^{-1}$ and accuracy of ±100 $\mu$mol mol$^{-1}$ in the range 0-1000 $\mu$mol mol$^{-1}$. The detection limit and accuracy are much lower than that for the LGR FMEA and may limit application of this sensor for detecting small emissions. These two sensors were examined here for illustrative purposes only, to demonstrate the higher performance of higher cost gas analysers and the improved data quality."

*Fig 1: Instrument labelled as uMEA*

Well spotted. This has been fixed and Fig. 1 now correctly labels the FMEA.

*L114: The FMEA may sample at 1Hz, but what is the turnover time of the cell and therefore what is the true sampling resolution of the system?*

We are unable to measure this currently as the LGR FMEA is requiring service due to a suspected failure of the solid state drive. However, viewing previous data where we have injected standard references gases with known $CH_4$ concentration (20, 50, and 100 ppm) into the LGR FMEA, it has taken approximately 30 seconds to return to ambient concentration after the gas has been removed. The below plot shows the LGR FMEA response following injection of ~50 ppm $CH_4$ on 30/03/2022 (*x* axis is time in seconds, *y* axis is the $CH_4$ mole fraction (dry) in ppm).

[Figure]

The below plot shows the LGR FMEA response following injection of ~100 ppm $CH_4$ on 21/05/2025 ($x$ axis is time in seconds, $y$ axis is the $CH_4$ mole fraction (dry) in ppm). This indicates that the cell turnover time is unlikely to change even over several years of operation.

[Figure]

In addition, Vogel et al. (2024) report a cell turnover time of 1-3 s for an LGR ultraportable greenhouse gas analyser (UGGA; a similar, more portable analogue of the rackmounted FMEA measuring $CH_4$ and $CO_2$ as opposed to $CH_4$ and $C_2H_6$). France et al. (2021), however, report a cell flushing (turnover) time of >10 s for another LGR UGGA.

A sentence was added to the Methodology section:

"The time taken for gas to be flushed through the optical cell (cell turnover time) was ~20 s; this turnover time is similar to those reported for analogous but smaller gas analysers (e.g., France et al., 2021)."

References

Vogel, F., Ars, S., Wunch, D., Lavoie, J., Gillespie, L., Maazallahi, H., Röckmann, T., Necki, J., Bartyzel, J., Jagoda, P., Lowry, D., France, J., Fernandez, J., Bakkaloglu, S., Fisher, R., Lanoiselle, M., Chen, H., Oudshoorn, M., Yver-Kwok, C., Defratyka, S., Morgui, J. A., Estruch, C., Curcoll, R., Grossi, C., Chen, J., Dietrich, F., Forstmaier, A., van der Gon, H. A. C. D., Dellaert, S. N. C., Salo, J., Corbu, M., Iancu, S. S., Tudor, A. S., Scarlat, A. I. and Calcan, A.: Ground-based

mobile measurements to track urbane methane emissions from natural gas in 12 cities across eight countries, Env. Sci. Tech., 58, 5, 2271-2281, https://doi.org/10.1021/acs.est.3c03160, 2024.

France, J. L., Bateson, P., Dominutti, P., Allen, G., Andrews, S., Bauguitte, S., Coleman, M., Lachlan-Cope, T., Fisher, R. E., Huang, L., Jones, A. E., Lee, J., Lowry, D., Pitt, J., Purvis, R., Pyle, J., Shaw, J., Warwick, N., Weiss, A., Wilde, S., Witherstone, J. and Young, S.: Facility level measurement of offshore oil and gas installations from a medium-sized airborne platform: method development for quantification and source identification of methane emissions, Atmos. Meas. Tech., 14, 71-88, https://doi.org/10.5194/amt-14-71-2021, 2021.

*L125: Expand this section so that the reader can understand the lags and data invalidation periods. 60s seems like a very long time to clear a portion of line if the lines to each node are being continually pumped. I'd like to see some of this analysis in the main paper.*

A 60 second lag time might seem quite long but given the identified cell turnover time of ~30 seconds (see above), this allows for approximately two full flushes of the optical cell. This ensures that there will be little-to-no contamination of gas from alternate sampling locations.

*L129: Is this really sufficient for traceability? I think this quality of calibration is quite poor and I would have expected better QC on this. Can this be replicated in the lab now with more cal gases to demonstrate that instrument performance is as expected over the measurement range seen.*

We agree with the reviewer that the use of a single-point reference gas standard is not ideal as we cannot rule out a non-linear response to $CH_4$ concentration. However, we do not refer to this as a "calibration" in the manuscript. It should also be noted that the LGR manufacturer reports an "extremely wide linear measurement range" (up to 100 ppm) for the GLA331 series of analysers. We do have some additional "calibration" data for the FMEA from 21/05/2025 where three standard reference gases (2 ppm, 20 ppm, 100 ppm) were sampled for approximately 5 minutes each with mean readings of 2.0069 ± 0.0006, 20.660 ± 0.002, and 98.536 ± 0.012 ppm respectively. These readings imply a small non-linear response, potentially as concentration approaches 100 ppm. It should be noted that these standard reference gas checks were performed following a service of the LGR FMEA so the data may not be directly relevant for correcting data reported in the main manuscript (prior to the service).

Unfortunately, as the LGR FMEA is requiring another maintenance service, it is not available for further calibration at this point.

We already recognised the limitation of a single-point calibration in the text (see below):

"In practice, it would be preferential to employ a calibration scheme with more than one single standard reference point to account for any non-linearity in analyser response across a range of $CH_4$ mole fractions. However, the consistency between the analyser response and the reference gas standard provides enough confidence that the data was of high precision for the duration of this field validation campaign."

*L169: I do have questions around wind persistence and whether it would become less of a controlling problem if you weren't having to average everything to hourly timesteps. I suspect that there isn't sufficient data to dig into this further, but I would consider looking at minute by minute enhanced methane readings and correlating with wind direction to see if that gives better data for locating an emission source.*

This is highly likely and we are in agreement that smaller averaging periods would reduce the impact of wind persistence and wind variability (as the wind direction is less likely to vary considerably over smaller time periods). Unfortunately, both Airviro and WindTrax do not deal well with missing data. This was the main reason for hourly averages – it allows for a concentration to be input to the models for each sampling location for each timestamp. Minute-average data would have only a concentration value for one location for each timestamp. Airviro and WindTrax are more likely to return errors (or fall over entirely) if there are missing values for more than one sampling location across a dataset.

There may be ways around this, potentially by removing/ignoring time stamps and treating data as time agnostic. This will be examined in the future.

*L200: As this is stated to be a validation study, I would like to see more information and justification around the design of the experiment – especially given that it is stated that this system will be used for landfill quantification, among other things.*

As mentioned above, we have removed the specific references to "validation" and rewritten to "assessment" instead as we believe this to be a fairer representation of the work done.

We have added additional text to the Meteorology section as below:

"A release height of ~2 m may be broadly representative of some low-level elevated sources but not of higher elevated sources such as stacks or chimneys. Similarly, the elevated release height (and the point source) is unlikely to be representative of landfill emission sources where methane is typically emitted through the subsurface across a wide area. The release scenario used here was chosen to be reasonably simple to conduct a first assessment of the FEDS system."

*Fig 3. Can the wind direction be plotted as dots so that there is no 360 jumps in the data.*

The meteorological and atmospheric data in this manuscript were plotted using openair, a freely available R package (https://openair-project.github.io/book/). There does not appear to be functionality to plot the wind direction as points (rather than a continuous line) within the openair package (to the best of the authors' knowledge). There is functionality to change the line type to dashed or dots but this does not remove the large "jumps" in data when wind direction moves across the 360° threshold. There is, however, functionality to plot the data as vertical bars (like a bar plot). Using image editing software, we have adjusted vertical bars for wind direction to appear as data points in the below plot. If the reviewer/editor believe this to be acceptable, then we would be happy to update the figure in the main manuscript.

[Figure]

*L260 section: I'd like to see a probability of detection metric determined for the set up if at all possible.*

Bell et al. (2023) defined the probability of detection as the "fraction of controlled releases classified as True Positive". This is an exceptionally useful metric when there were many controlled releases (e.g., Bell et al. (2023) used several hundred). As this work only used four controlled release experiments, the probability of detection metric is likely less meaningful (compared to the hourly success index reported elsewhere in our manuscript). The probability of detection for Airviro was 75% (3 out of 4 controlled release detections) and for WindTrax was 100% (4 out of 4 detections).

We have added a new section to the manuscript comparing our results against other controlled release experiments and probability of detection is covered there. The section is too large to be copied here and should be referred to in the main manuscript revision submitted.

*Fig 4. This is not at all intuitive and I would give serious consideration to redefining how this is presented and instead look at methane excess over background as the primary metric rather than just atmospheric mole fraction as I think it would provide much more immediate understanding*

Does the confusion here lie with the fact that the wind rose plots emphasise the most abundant wind direction? We could potentially normalise the wind direction data paddles (so that they all display with the same radii) but this will only exaggerate the less abundant wind directions. We unfortunately do not see the advantage of subtracting the background and replotting. Firstly, assuming a constant background value (of e.g., 2 ppm), this would just redistribute the range of plotted concentration values. Secondly, this may introduce some element of bias if the background is not correctly accounted for – background calculation is difficult due to variability in different wind directions and over time. Finally, we could also remove (not plot) data below a certain threshold (e.g., 2.5 ppm) but this would leave the sampling locations with different amounts of data (with some likely having no data to plot).

At this point, we have decided to leave the plot as is unless there is further recommendations for improvement.

*L434: Reference issue*

Resolved.

*L458-462: Whilst these models may perform to the stated stats, I don't get a sense of understanding as to why they are performing well or poorly (with the exception of the discussion of the wind persistence). Is there more to the modelling than this / more nuance?*

We also examined a number of other parameters across all modelling options which are presented in the appendices. We chose to present these analyses outside of the main text to reduce the size of the main manuscript. We believe the key findings of the overall assessment are still presented in the manuscript.

*L497: Surely measuring more methane downwind from the source can't be considered a conclusion?*

We have removed this line of the conclusion.

However, this was the conclusion of another recent controlled release experiment with point sensor networks: "when emissions are directly upwind of a sensor, the mixing ratio readings differ when [controlled releases] are active [compared to when controlled releases are not active], indicating that a signal exists using sensor technology." (Day et al., 2024).

References

Day, R. E., Emerson, E., Bell, C. and Zimmerle, D.: Point sensor networks struggle to detect and quantify short controlled releases at oil and gas sites, Sensors, 24, 8, 2419, https://doi.org/10.3390/s24082419, 2024.

*L513: Give % errors so that we can see how performance was without having to know what the release rates were.*

We have added a bit of text to the sentence to divulge the controlled release rate.

"… (and for a controlled release rate of 1 kg h$^{-1}$)…"

*Conclusions: General – I'd suggest shortening and tightening up once the major corrections are sorted. It is a little long and unconcise as it stands.*

The length of the conclusions has been significantly shortened by moving some text to the new section comparing the FEDS system performance against other continuous monitoring solutions in the literature.

Reviewer 2

*The paper describes the validation of a monitoring system to be used to quantify methane emissions over long-term periods from large, aerodynamically complex sources (e.g. natural gas network, landfill, and waste treatment sites). The validation was conducted using four controlled emissions (point source; 20-hour duration; fixed emission rate 1 kg CH4 h-1) with the methane mixing ratio measurements being made a short distance (~35 m) downwind of the release over a flat grass fetch. The measurement campaign lasted 18-days in March/April 2022. Results show good agreement between calculated emissions and the controlled emission rate.*

*The paper is written well but similar results have been published by other studies, research in this field has moved on considerably.*

We thank the reviewer for their review and constructive criticisms. Responses to general comments and suggestions can be found below.

*Firstly, there is little novelty in the work. Many systems have been developed that can quantify methane emissions from point sources a relatively small distance away over an aerodynamically simple wind field. New research in this field either investigates using low-cost/low-power technology that can be deployed remotely or novel dispersion modelling approaches. The authors claim the USP of this paper is that it uses a single expensive analyzer connected to multiple sampling locations. However, methane analyzers, such as the one used here (no details were given on the spec and I cannot find it on ABB's website), are expensive (USD 20,000 - $50,000), require mains power to operate and a controlled climate environment to operate in. Moreover, the methane mole fraction data were used to generate an hourly average – raising the question of whether a trace methane analyzer is actually needed. As a result, those quantifying methane emissions to justify GHG emission estimates (e.g. landfill and O&G operators) are using alternative instruments that are lower cost, lower power and can be adapted to operate in a range of environmental conditions. The data are presented but there is no synthesis or research questions asked other than - does this approach work in the simplest of emissions scenarios.*

The analyser used in this work was an ABB Ltd. Los Gatos Research Inc. (LGR) Fast Methane-Ethane Analyzer (FMEA). This was mentioned in the Methodology section and in the caption on Figure 1. We have added additional information to the text that the analyser was one of LGR's GLA331 series although it appears that this exact analyser may no longer be available on ABB's website. We have added a table to the Appendices (A1) which includes the manufacturer-reported specifications for this analyser.

We also deployed low-cost sensors (Figaro TGS2611) at each sampling location (alongside the periodic measurement at each location by the LGR FMEA) as a comparison. We observed a strong temperature and humidity dependency of the low-cost sensor results with substantial inter-sensor variability in these relationships. The concentrations derived from the low-cost sensors were generally not in agreement with co-located LGR FMEA data (when adjusted for time response) and the low-cost sensors routinely reported concentrations below the atmospheric methane background. The low-cost sensors were calibrated in the laboratory (but not whilst in the field) and exhibited some drift (again, the extent of drift differed between sensors) whilst deployed as part of this field assessment.

Whilst the reviewer might be correct that some landfill and O&G operators typically use lower cost, lower power, and more adaptable sensors, there are serious metrological concerns about both the accuracy of these options and their representability over long periods. For example, landfill operators in the UK are only required to undertake annual walkover surveys of surface emissions using a relatively low-cost sensor (e.g., flame ionisation detector (FID)) (LFTGN07 v2, 2010). For comparison, a commercially available FID instrument (Gas-Tec) provides (operator-reported) accuracy to ±10% - an order of magnitude worse than the (operator-reported) accuracy of the analyser used in this work. Other sensors, such as the Gazomat CATEX 3, have minimum detection limits of 50 ppm and accuracy of ±100 ppm in the range of 0-1000 ppm. We are aware that commercial measurement services may well be looking into the many different low-cost sensors which are now available on the market, or at using optical gas imaging. Whilst there are benefits to using these low-cost systems, there are multiple issues with all these approaches which raise some questions (listed below).

1. The suitability of a lower performance sensor with potentially higher biases and uncertainties, potential for response to interfering analytes which are not the measurand, and reduced quality of resultant concentration data.
2. Representability of annualised surveys or spatially discrete surveys such as flux boxes.
3. The safety of conducting walkover surveys with personnel in complex environments with potentially hazardous ATEX areas.
4. Open question as to the capability of deriving accurate emission estimates from some of these measurement options (such as optical gas imaging).

The authors believe it is unfair to state that the market has moved on to lower cost options when there is still a clear need to make high quality measurements with metrological traceability – as FEDS is trying to provide. Indeed, the merits of multiple approaches working together across scales (temporal, spatial, economical) must not be understated. In fact, much recent work published in AMT has used high-cost gas analysers for fence-line monitoring (Mbua et al., 2025) or for other ground-based uses (e.g., Dubey et al., 2025; Follansbee et al., 2025; Liu et al., 2025; Wietzel et al., 2025) including specifically LGR analysers (Joo et al., 2025). The UK's tall tower network (Deriving Emissions linked to Climate Change; DECC: Stanley et al., 2018) also uses these state-of-the-art analysers with robust calibration to ensure high precision and metrologically traceable data.

NPL are also actively involved in multiple projects with the UK Government and across Europe, working with colleagues in academia, regulation, and inventory reporting, which all make use of these higher cost sensors to collect high quality data. Very few of these projects (where accuracy and metrological suitability are prioritised) are using lower-cost sensors (at this point in time). That isn't to say that low-cost sensors have no place in emission detection (or will never have a place) but their outputs are simply not trusted at the moment, and they require further assessment to be trusted going forwards.

Practically, a system such as the FEDS, which is intended for site- or facility-scale emission quantification, would utilise some combination of low-cost sensors with high precision gas analysers. The low-cost sensors would provide temporally continuous data from multiple sampling locations, potentially at reduced quality and only once the reliability of low-cost sensors improves. The gas analyser would provide high precision and metrologically traceable (via robust calibration) measurements of a subsample of locations, which could be routinely compared against the low-cost sensor data to ensure consistency across all sensors. Currently, low-cost sensors provide utility for leak detection but are potentially of less use for producing high quality measurements of methane concentration for emission quantification purposes.

References

Dubey, M. L., Santos, A., Moyes, A. B., Reichl, K., Lee, J. E., Dubey, M. K., LeYhuelic, C., Variano, E., Follansbee, E., Chow, F. K. and Biraud, S. C.: Development of a forced advection sampling technique (FAST) for quantification of methane emissions from orphaned wells, Atmos. Meas. Tech., 18, 2987-3007, https://doi.org/10.5194/amt-18-2987-2025, 2025.

Follansbee, E., Lee, J. E., Dubey, M. L., Dooley, J. F., Shuck, C., Minschwaner, K., Santos, A., Biraud, S. C. and Dubey, M. K.: Orphasned oil and gas well methane emission rates quantified using Gaussian plume inversions of ambient observations, Atmos. Meas. Tech., 18, 4527-4542, https://doi.org/10.5194/amt-18-4527-2025, 2025.

LFTGN07 v2, Guidance on monitoring landfill gas surface emissions, Environment Agency, 2010.

Joo, J., Jeong, S., Lee, H., Kim, Y., Shin, J., Kim, D. and Chang, D.: Methane quantification of LNG gas-fired power plant in Seoul, South Korea, EGUSPhere, https://doi.org/10.5194/egusphere-2025-4379, 2025.

Liu, Y., Miles, N. L., Richardson, S. J., Barkley, Z. R., Miller, D. O., Kofler, J., Handley, P., DeVogel, S. and Davis, K. J.: Laboratory and field assessment of mid-infrared absorption (MIRA) instrument performance for methane and ethane dry mole fractions, EGUSphere, https://doi.org/10.5194/egusphere-2025-4950, 2025.

Mbua, M., Riddick, S. N., Kiplimo, E., Shonkwiler, K. B., Hodshire, A. and Zimmerle, D.: Evaluating the feasibility of using downwind methods to quantify point source oil and gas emissions using continuously monitoring fence-line sensors, Atmos. Meas. Tech., 18, 5687-5703, https://doi.org/10.5194/amt-18-5687-2025, 2025.

Stanley, K. M., Grant, A., O'Doherty, S., Young, D., Manning, A. J., Stavert, A. R., Spain, T. G., Salameh, P. K., Harth, C. M., Simmonds, P. G., Sturges, W. T., Oram, D. E. and Derwent, R. G.: Greenhouse gas measurements from a UK network of tall towers: technical description and first results, Atmos. Meas. Tech., 11, 1437-1458, https://doi.org/10.5194/amt-11-1437-2018, 2018.

Wietzel, J. B., Korben, P., Hoheisel, A. and Schmidt, M.: Best practices and uncertainties in $CH_4$ emission quantification: employing mobile measurements and Gaussian plume modelling at a biogas plant, Atmos. Meas. Tech., 18, 4631-4645, https://doi.org/10.5194/amt-18-4631-2025, 2025.

*Secondly, claims made that the FEDS could be used to quantify emissions of gases from other sources are not validated by the data presented in the paper. The controlled release experiments are very basic when compared against emission scenarios generated at FluxLab (https://fluxlab.ca/) or METEC (https://metec.colostate.edu/). The authors do not describe why they have chosen a single, continuous point source emission of 1 kg CH4 h-1, released at ~2 m above the ground for 20 hours or what real-world emission scenario this is meant to represent. This emission not representative of those from either landfills (heterogenous, area source emission) or wastewater treatment sites (both point and area source emissions). This could simulate a source on the natural gas network (usually point source), but in real-life emissions are typically intermittent (e.g. cycling with separator dumps). The aerodynamic complexity of the experiment is a real weakness, and I find it difficult to imagine any real-world emissions scenario that this would represent.*

Reviewer 1 also noted concerns surrounding the use of the word "validation", and instances of this word have now been removed throughout the manuscript and replaced with "assessment" (including in the submission title).

The authors accept that the controlled release experiments conducted at facilities such as METEC or TADI, or those conducted by FluxLab, are much more complex than the ones presented in this study. Those facilities have capability for multiple tests conducted over a wide range of release rates and using equipment intended to directly mimic real-world emissions. However, the authors would argue that such facilities are very unique, incredibly expensive both to set up and run, and such experiments require vast amounts of gas. The UK does not have any permanent facility capable of providing controlled releases. As such, NPL's Controlled Release Facility, whilst simple, is the best available facility in the UK for providing metrologically traceable releases of trace gas species. NPL's CRF is also mobile (portable) which means it can be used in a range of environments.

However, we have attempted another assessment of the FEDS system with greater ambition; eight controlled release scenarios with differing durations and differing release heights. However, by trying to test multiple different scenarios in a short space of time we ended up with insufficient data to provide a robust assessment and quality assurance of the FEDS method from this dataset. There were also issues with some meteorological sensors deployed during this assessment. The submitted manuscript represents the second attempt, where a concerted effort was made to ensure enough data was collected to provide a robust analysis of the FEDS approach (especially from a modelling perspective), albeit when using only a very simplistic emission scenario. We acknowledge that the scope of the assessment was limited (largely due to a restricted budget) but this was the extent of the data collected.

We do hope to expand controlled release testing of the FEDS system and other methods in the future, but we require further funding to do so.

We would also like to point out that other methods have also been tested using relatively simple suites of controlled release experiments. For example, Shah et al. (2019) measured eight controlled releases of $CH_4$ at two different release rates (~5 and ~10 kg/h) for their near-field Gaussian method. Riddick et al. (2022) used three controlled releases of $CH_4$ (0.08, 0.17, and 0.31 kg/h; constant release rates for 48 h) to characterise the capability of some low-cost methane sensors for detecting emissions. Dubey et al. (2025) tested a method for quantifying emissions from orphaned gas wells using six controlled releases (rates ranged from 0.001 to 0.04 kg/h; each release lasted ~30 mins; release height 1 m above ground).

References

Shah, A., Allen, G., Pitt, J. R., Ricketts, H., Williams, P. I., Helmore, J., Finlayson, A., Robinson, R., Kabbabe, K., Hollingsworth, P., Rees-White, T. C., Beaven, R., Scheutz, C. and Bourn, M.: A near-field Gaussian plume inversion flux quantification method, applied to unmanned aerial vehicle sampling, Atmosphere, 10, 7, 396, https://doi.org/10.3390/atmos10070396, 2019.

Riddick, S. N., Ancona, R., Cheptonui, F., Bell, C. S., Duggan, A., Bennett, K. E. and Zimmerle, D. J.: A cautionary report of calculating methane emissions using low-cost fence-line sensors, Elem. Sci. Anth., 10, 1, 00021, https://doi.org/10.1525/elementa.2022.00021, 2022.

Dubey, M. L., Santos, A., Moyes, A. B., Reichl, K., Lee, J. E., Dubey, M. K., LeYhuelic, C., Variano, E., Follansbee, E., Chow, F. K. and Biraud, S. C.: Development of a forced advection sampling technique (FAST) for quantification of methane emissions from orphaned

wells, Atmos. Meas. Tech., 18, 2987-3007, https://doi.org/10.5194/amt-18-2987-2025, 2025.

*Overall, I would recommend: 1. that the authors reanalyze their data to present something more novel (e.g. down-sampling the mole fraction data to assess the highest sensor detection limits that still yields the same results – how does this affect the cost of the system and are there available sensors that could be used instead of the LGR?); and 2. be very clear in the conclusions what the limitations of these validation experiments are and place the controlled emission scenario you have used in a realistic, real-world context.*

We believe the work presented to be a description and assessment of a novel system (single analyser, multi-point sampling network) for long-term continuous emission monitoring. The novelty of the system lies in the metrological traceability inherent to a continuously sampling network that is less easily achieved with low-cost sensors with their distinct lack of capacity for easy calibration. We recognise that there are certain improvements that could be realised with the FEDS system and some of these have been actioned in work following this initial assessment. We also recognise that the controlled release experiments could be greatly expanded. Where this work provides novelty over that provided by larger controlled release experiments at METEC (or similar), is in a more detailed description of the continuous monitoring system, reported processes for data analysis and quality assurance, and evaluation of the impact of modelling variables on results. This contrasts to e.g., Bell et al. (2023) which did not provide much detail of the commercial continuous monitoring solutions assessed (e.g., the sensors used, sampling approaches, flux modelling approach) to preserve commercial anonymity.

There are, of course, alternative sensors that could be used instead of an LGR. Aeris, Aerodyne, and Picarro all manufacture gas analysers for atmospheric $CH_4$ sampling that are in use by the research community across a number of different systems (e.g., long-term stationary sampling for background assessment, mobile sampling using vehicles, mobile sampling using UAVs, the tracer gas dispersion method etc.). Whilst we understand the reviewers concerns that the cost of these analysers is high, there are simply too many metrological challenges with lower-cost sensors to justify their unaccompanied use for accurate emissions monitoring at their current technological readiness level.

We have added the following text to the Conclusions (in addition to some other text detailed elsewhere):

"The assessment approach used a limited set of controlled release experiments of $CH_4$ which did not replicate the range of parameters present for real-world emission sources. Controlled release experiments should be greatly expanded to fully validate the FEDS system for real-world emission scenarios from simulated oil and gas facilities or landfill-type (area) emission sources. For example, controlled releases in this work were conducted over reasonably long periods of time (>20 hours) and hence the performance of FEDS for transient emission events of considerably shorter duration (i.e., hours or minutes) was not assessed. However, we provide data and evidence for the quality assurance of quantified emissions output by local scale reverse dispersion models, largely through interpretation of the wind field."